# Smooth Activations and Reproducibility in Deep Networks

## Abstract

Deep networks are gradually penetrating almost every domain in our lives due to their amazing success. However, with substantive performance accuracy improvements comes the price of *irreproducibility*. Two identical models, trained on the exact same training dataset may exhibit large differences in predictions on individual examples even when average accuracy is similar, especially when trained on highly distributed parallel systems. The popular Rectified Linear Unit (ReLU) activation has been key to recent success of deep networks. We demonstrate, however, that ReLU is also a catalyzer to irreproducibility in deep networks. We show that not only can activations smoother than ReLU provide better accuracy, but they can also provide better accuracy-reproducibility tradeoffs. We propose a new family of activations; Smooth ReLU (*SmeLU*), designed to give such better tradeoffs, while also keeping the mathematical expression simple, and thus implementation cheap. SmeLU is monotonic, mimics ReLU, while providing continuous gradients, yielding better reproducibility. We generalize SmeLU to give even more flexibility and then demonstrate that SmeLU and its generalized form are special cases of a more general methodology of REctified Smooth Continuous Unit (RESCU) activations. Empirical results demonstrate the superior accuracy-reproducibility tradeoffs with smooth activations, SmeLU in particular.

## 1 Introduction

Recent developments in deep learning leave no question about the advantages of deep networks over classical methods, which relied heavily on linear convex optimization solutions. With their astonishing unprecedented success, deep models are providing solutions to a continuously increasing number of domains in our lives. These solutions, however, while much more accurate than their convex counterparts, are usually *irreproducible* in the predictions they provide. While average accuracy of deep models on some validation dataset is usually much higher than that of linear convex models, predictions on individual examples of two models, that were trained to be identical, may diverge substantially, exhibiting *Prediction Differences* that may be as high as non-negligible fractions of the actual predictions (see, e.g., Chen et al. (2020); Dusenberry et al. (2020)). Deep networks express (only) what they learned. Like humans, they may establish different beliefs as function of the order in which they had seen training data (Achille et al., 2017; Bengio et al., 2009). Due to the huge amounts of data required to train such models, enforcing determinism (Nagarajan et al., 2018) may not be an option. Deep networks may be trained on highly distributed, parallelized systems. Thus two supposedly identical models, with the same architecture, parameters, training algorithm and training hyper-parameters that are trained on the same training dataset, even if they are initialized identically, will exhibit some randomness in the order in which they see the training set and apply updates. Due to the highly non-convex objective, such models may converge to different optima, which may exhibit equal average objective, but can provide very different predictions to individual examples. Irreproducibility in deep models is not the classical type of epistemic uncertainty, widely studied in the literature, nor is it overfitting. It differs from these phenomena in several ways: It does not diminish with more training examples like classical epistemic uncertainty, and it does not cause degradation to test accuracy by overfitting unseen data to the training examples.

While irreproducibilty may be acceptable for some applications, it can be very detrimental in applications, such as medical ones, where two different diagnoses to the same symptoms may be unacceptable. Furthermore, in online and/or re-enforcement systems, which rely on their predictions to

determine actions, that in turn, determine the remaining training examples, even small initial irreproducibility can cause large divergence of models that are supposed to be identical. One example is sponsored advertisement online Click-Through-Rate (CTR) prediction (McMahan et al., 2013). The effect of irreproducibility in CTR prediction can go far beyond changing the predicted CTR of an example, as it may affect actions that take place downstream in a complex system. Reproducibility is a problem even if one trains only a single model, as it may be impossible to determine whether the trained model provides acceptable solutions for applications that cannot tolerate unacceptable ones.

A major factor to the unprecedented success of deep networks in recent years has been the *Rectified Linear Unit (ReLU)* activation, (Nair & Hinton, 2010). ReLU nonlinearity together with back-propagation give simple updates, accompanied with superior accuracy. ReLU thus became the undisputed activation used in deep learning. However, is ReLU really the best to use? While it gives better optima than those achieved with simple convex models, it imposes an extremely non-convex objective surface with many such optima. The direction of a gradient update with a gradient based optimizer is determined by the specific example which generates the update. Thus the order of seeing examples or applying updates can determine which optimum is reached. Many such optima, as imposed by ReLU, thus provide a recipe for irreproducibility. In recent years, different works started challenging the dominance of ReLU, exploring alternatives. Overviews of various activations were reported in Nwankpa et al. (2018); Pedamonti (2018). Variations on ReLU were studied in Jin et al. (2015). Activations like SoftPlus (Zheng et al., 2015), *Exponential Linear Unit (ELU)* (Clevert et al., 2015), *Scaled Exponential Linear Unit (SELU)* (Klambauer et al., 2017; Sakketou & Ampazis, 2019; Wang et al., 2017), or *Continuously differentiable Exponential Linear Unit (CELU)* (Barron, 2017) were proposed, as well as the *Gaussian Error Linear Unit (GELU)* (Hendrycks & Gimpel, 2016). Specifically, the *Swish* activation (Ramachandran et al., 2017) (that can approximate GELU) was found through automated search to achieve superior accuracy to ReLU. Further activations with similarity to GELU were proposed recently; *Mish* (Misra, 2019), and *TanhExp* (Liu & Di, 2020). Unlike ReLU, many of these activations are *smooth* with continuous gradient. Good properties of smooth activations were studied as early as Mhaskar (1997) (see also Du (2019); Lokhande et al. (2020)). These series of papers started suggesting that smooth activations, if configured properly, may be superior to ReLU in accuracy. Recent work by Xie et al. (2020), that was done subsequently to our work, and was inspired from our results that we report in this paper, (Lin & Shamir, 2019), demonstrated also the advantage of smooth activations for adversarial training.

**Our Contributions:** In this paper, we first demonstrate the advantages of smooth activations to reproducibility in deep networks. We show that not only can smooth activations improve accuracy of deep networks, they can also achieve superior tradeoffs between reproducibility and accuracy, by attaining a lower average *Prediction Difference (PD)* for the same or better accuracy.

Smooth activations, like Swish, GELU, Mish, and TanhExp, all have a very similar non-monotonic form, that does not provide a clear stop region (with strict 0; not only approaching 0), and slope 1 region. While these activations approximate the mathematical form of ReLU, they lack these properties of ReLU. All these activations including SoftPlus also require more expensive mathematical expressions, involving exponents, and, in some, logarithms, or even numerically computed values (e.g., GELU). This can make deployment harder, especially with certain simplified hardware that supports only a limited number of operations, as well as can slow down training due to the heavier computations. Unlike ReLU, which can be transformed into *Leaky* ReLU, the smooth activations described cannot be easily transformed into more general forms. In this work, we propose *Smooth ReLU (SmeLU)*, which is mathematically simple and based only on linear and quadratic expressions. It can be more easily deployed with limited hardware, and can provide faster training when hardware is limited. SmeLU provides a clearly defined 0 activation region, as well as a slope 1 region, is monotonic, and is also extendable to a leaky or more general form. SmeLU gives the good properties of smooth activations providing better reproducibility as well as better accuracy-reproducibility tradeoffs. Its generalized form allows even further accuracy improvements. The methodology to construct SmeLU is shown to be even more general, allowing for more complex smooth activations, which are all clustered under the category of *Rectified Smooth Continuous Units (RESCUs)*.

**Related Work:** Ensembles (Dietterich, 2000) have been used to reduce uncertainty (Lakshminarayanan et al., 2017). They are useful also for reducing irreproducibility. However, they make models more complex, and can trade off accuracy in favor of reproducibility if one attempts to keep constant computation costs (which require reducing capacity of each component of the ensemble). Compression of deep networks into smaller networks that attempt to describe the same information

is the emerging area of *distillation* (Hinton et al., 2015). Predictions of a strong *teacher* train a weaker *student* model. The student is then deployed. This approach is very common if there are ample training resources, but deployment is limited, as for mobile network devices. *Co-distillation*, proposed by Anil et al. (2018) (see also Zhang et al. (2018)), took advantage of distillation to address irreproducibility. Instead of unidirectional transfer of knowledge, several models distill information between each other. They attempt to converge to the same solution. The method requires more training resources to co-train models, but deployment only requires a single model. A somewhat opposite approach; *Anti-Distillation*, to address irreproducibility was proposed by Shamir & Coviello (2020), embracing ensembles, with an additional loss that forces their components away from one another. Each component is forced to capture a (more) different part of the objective space, and as a whole, the predictions of the ensemble are more reproducible. To the best of our knowledge, all previously reported techniques to address irreproducibility in deep networks required some form of ensembles. In this work, we leverage smooth activations, and do not require ensembles.

**Outline:** Section 2 proposes several PD metrics, which we use to measure irreproducibility. Next, we overview our setup and smooth activations in Section 3, and describe SmeLU and its generalization in Section 4. Experimental results are shown in Section 5.

## 2 PREDICTION DIFFERENCE

Average individual per-example *Prediction Difference (PD)* over a set of models that are configured, trained, and supposed to be identical over some validation dataset can be defined in various ways. We refer to Shamir & Coviello (2020) for a more detailed discussion. We opt to the same definitions they used, where we describe the classification case, in which we measure the PD using some $L_p$ norm on the actual label predictions. Following their notation, denote the number of models by $M$, and the number of validation examples by $N$. Let $P_{n,m}$ be the distribution over labels predicted by model $m$ for example $n$. ($P_{n,m}(\ell)$ is the probability predicted for label $\ell$.) Let $P_n \overset{\triangle}{=} \sum_m P_{n,m}/M$ be the expected distribution over labels over all $M$ models. Then, the $p$th norm PD, $\Delta_p$, is given by

$$\Delta_p = \frac{1}{N} \sum_{n=1}^{N} \cdot \frac{1}{M} \sum_{m=1}^{M} \|P_{n,m} - P_n\|_p = \frac{1}{N} \sum_{n=1}^{N} \cdot \frac{1}{M} \sum_{m=1}^{M} \cdot \left[ \sum_{\ell} |P_{n,m}(\ell) - P_n(\ell)|^p \right]^{1/p}. \quad (1)$$

With practical large scale systems, with very high training costs, we can use $M = 2$, where for binary labels, $\Delta_1 = 1/N \cdot \sum_n |P_{n,1}(1) - P_{n,2}(1)|$, where 1 is the positive label. As in Shamir & Coviello (2020), we can consider *relative PD*, $\Delta_1^r$, normalizing the innermost summand in (1) by $P_n(\ell)$, which for binary problems can be tweaked into $\tilde{\Delta}_1^r$, normalizing by $P_n(1)$ instead. PD, $\Delta_1^L$, can be computed only on the observed true label, by replacing the innermost sum on $\ell$ in (1) by $|P_{n,m}(\ell_{true}) - P_n(\ell_{true})|$ for the true label, normalizing by $P_n(\ell_{true})$. Its computation, however, requires knowledge of the true label. Finally, in classification, one can use *Hamming PD*; $\Delta^H$, specifying the average fraction of labels predicted differently between model pairs.

## 3 SMOOTH ACTIVATIONS

We consider standard Neural Networks. In addition to numerical features, inputs can also be embedding vectors representing unmeasurable features, learned together with weights and biases of hidden layers. Networks can be deep or shallow (a few or even a single hidden layer). The output of the matrix multiplication in any of the networks' hidden layers is activated by a nonlinear activation. Our results apply in a standard setup, where training passes over a training dataset for multiple epochs, allowing example revisits. We also consider the *online* setting, where training consists of a single pass over the data. Inference is applied to unseen examples before updates. Normally, updates are applied in mini-batches. Some form of normalization may be applied in the hidden layers to limit signal range, guarding from vanishing or exploding gradients, and providing some scale invariance. Without normalization, activations can start drifting to become larger. As we show, smooth activations are controlled by parameters that dictate their behavior. Inputs that scale up generate an effect of scaling down widths of the activation regions. As we will observe, slower smoother gradient changes, manifested as wider input regions for the same output change, are more reproducible. With such drifts, they gradually appear to be narrower, slowly losing reproducibility benefits. Normalizations, as weight normalization (Salimans & Kingma, 2016), layer normalization (Ba et al., 2016),

or batch normalization (Ioffe & Szegedy, 2015) can be considered. In this work, however, we use a slightly different form of weight normalization, where some $L_p$ norm normalizes all matrix link weights incoming to an output neuron of a layer. We specifically normalize the $L_2$ norm of the incoming matrix weights into the neuron to some norm $v$ without centering around the mean.

Let $W^\ell$ be the weight matrix incoming into layer $\ell$ from layer $\ell - 1$, use $\mathbf{w}_j^\ell$ to denote the $j$th row of $W^\ell$. Then, the pre-activation neuron vector of layer $\ell$ is given by $\mathbf{a}^\ell = W^\ell \cdot f(\mathbf{a}^{\ell-1})$, where $f(\cdot)$ denotes the activation non-linearity (except for the input layer $\ell = 0$, where it is the identity). Then, with $L_2$ weight normalization, we replace $\mathbf{w}_j^\ell$ by $\tilde{\mathbf{w}}_j^\ell \triangleq v \cdot \mathbf{w}_j^\ell / \|\mathbf{w}_j^\ell\|_2$. Norms, other than $L_2$, as well as other normalization regimes can also be used. As we demonstrate below, for smooth activations, the norm $v$ can trade off with the activation parameters, tuning the accuracy/reproducibility tradeoffs, but not changing the good effects of smooth activations. As demonstrated in Appendix A, lack of normalization with nonlinearity in the form of clipping values can diminish the benefits of smooth activations. Note that recent work by Liu et al. (2020) combined normalization with activations.

As described, several forms of smooth activations have been recently proposed in the literature. They all attempt to mimic the shape of ReLU but with a smoother function. They also try to avoid similarity to the smooth Sigmoid, $\sigma(x) = 1/[1 + \exp(-x)]$, which was considered several decades ago, but had limited success due to diminishing gradients. Many of them can be parameterized by one parameter or more. While some are smooth for all parameter values, others, like SELU, are only smooth with specific values of the parameters. Let $\beta$ be the main activation parameter (SELU will also have another parameter $\lambda$). Let erf$(\cdot)$ denote the standard error function and $\Phi(\cdot)$ the standard normal CDF. Then, activation functions for some smooth activations are given by

$$y_{\text{SELU}} = \lambda \cdot \begin{cases} x, & \text{if } x > 0 \\ \beta e^x - \beta, & \text{if } x \le 0 \end{cases} \tag{2}$$

$$y_{\text{SoftPlus}} = \frac{1}{\beta} \cdot \log[1 + \exp(\beta \cdot x)] \tag{3}$$

$$y_{\text{Swish}} = x \cdot \sigma(\beta \cdot x) \tag{4}$$

$$y_{\text{GELU}} = x \cdot \frac{1}{2} \cdot \left[1 + \text{erf}\left(\beta \cdot x / \sqrt{2}\right)\right] = x \cdot \Phi(\beta \cdot x) \tag{5}$$

$$y_{\text{Mish}} = x \cdot \tanh[\log(1 + \exp(\beta \cdot x))] \tag{6}$$

$$y_{\text{TanhExp}} = x \cdot \tanh\left(e^{\beta x}\right). \tag{7}$$

ELU is SELU with $\lambda = 1$, and CELU is a differentially continuous variant of SELU, shown in Appendix C. We generalized SoftPlus, GELU, Mish, and TanhExp by adding the $\beta$ parameter. GELU can be approximated by Swish with $\text{GELU}_\beta(x) \approx x\sigma(\sqrt{8/\pi}\beta x)$. Fig. 1 shows the different activations and their gradients for different values of the parameter $\beta$. (More detailed figures are shown in Appendix C.) All the activations described thus far share a region in which the activation function limits changes in the signal with a gradient close to $0$ (on the left), and a region in which the gradient approaches $1$ on the right. The width of the middle transition region is a function of the parameter $\beta$, where it is wider with smaller $\beta$ and narrower with greater $\beta$. For $\beta \to \infty$, the activation resembles ReLU, and for $\beta \to 0$, it becomes closer to linear.

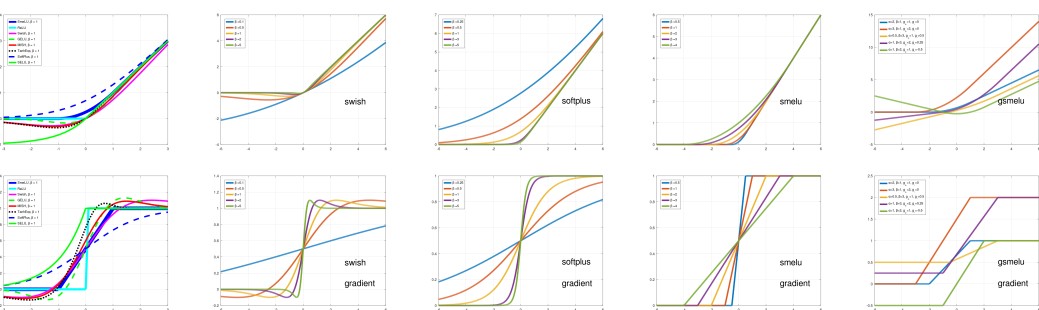

Figure 1: Smooth activations (top) and their gradients (bottom) for different $\beta$ parameter values.

The discontinuous gradient of the ReLU partitions the parameter space into regions, each of which has its own local optimum. Many of the local optima may be identical in the overall total objective, but not on how they predict for individual examples. Randomness in training leads the model into one of the regions, and locks it in a region of some local optima. Specifically, sudden jumps in the gradient, like those of ReLU, may lock some hidden units at some state, requiring other units to compensate and move towards their nearest optima. If this happens early in training (see, e.g., Achille et al. (2017)), it determines the trajectory of the optimization based on the examples seen and the updates applied. Different example and update orders thus lead to different trajectories.

Smoother activations give a training model less opportunities to diverge. Wider transition regions allow gradients to change more slowly, and units in the network to gradually adjust to updates and changes. Wider transition regions are closer in shape to more reproducible linear functions. However, a good activation must also retain good accuracy. The Sigmoid was unable to do that. Unfortunately, while reproducibility mostly tends to improve with widening the transition region of the activation, accuracy moves mostly in the opposite direction. As we lower $\beta$ and widen the region, we generally trade accuracy for reproducibility. This, however, is not the full picture. While moving towards ReLU improves accuracy on datasets we observed, the optimal point is not at the ReLU point of $\beta \to \infty$ but for some finite $\beta$, after which accuracy degrades. This shows that properly designed smooth activations for the specific dataset can be actually superior to ReLU in both accuracy and reproducibility. They also provide a knob to trade off between the two. Similar behavior is also observed in the opposite direction, where the best reproducibility for some datasets may be achieved for some $\beta > 0$, worsening for smaller values. The overall tradeoffs are dataset, model architecture and model configuration dependent.

## 4 SmeLU to the RESCU

Swish, GELU, Mish, and TanhExp, all have a non-monotonic region on the left of the transition region, neighboring the signal suppression region. This may be vulnerable to input irregularities, possibly less interpretable, and perhaps suboptimal. SoftPlus is monotonic, but the wider the transition region, the farther it is from the origin, degrading its accuracy. In addition, for all these activations, there are no clean stop and pass regions like in ReLU, except for very large $\beta$. Asymptotically, for $\beta \to \infty$, all tend to look like ReLU, but for wider transition regions, which are good for reproducibility, they do not reach the ReLU extremes on the left and right. Furthermore, all the activations considered require complex mathematical functions, such as exponents. Such functions require more complex hardware implementations, that could be very costly in huge scale systems. The computationally heavy functions could also slow down training and inference. We now show a very simple smooth ReLU activation, *SmeLU*, that benefits from smoothness, monotonicity, clean stop and pass regions, a very simple mathematical implementation, and comparable or better accuracy-reproducibility tradeoffs. We show that the same design is extendible to a generalization of this activation as well as to more sophisticated extensions.

**SmeLU:** In some literature, SoftPlus is referred to as Smoothed ReLU. Recent work (Bresler & Nagaraj, 2020) also considered smoothing ReLU using initial coefficients of its Fourier series. ReLU is piecewise continuous linear, but with a gradient discontinuity. As shown in Chen & Ho (2018); Montufar et al. (2014), the concept of piecewise linear can be extended to multiple pieces. We take a further step here and construct a piecewise continuous activation but also with a continuous gradient, by simply enforcing these conditions. The general idea is to define the activation as a piecewise linear and quadratic (resembling Huber loss (Huber, 1992) on one side). On both sides, we match ReLU and its gradients, and we fit a quadratic middle region between. We define $\beta$ to be the *half-width* of a symmetric transition region around $x = 0$. Note that this definition of SmeLU uses a parameter $\beta$ that is reciprocal to the $\beta$ in Swish and other activations like it in the sense that larger $\beta$ here gives a wider transition region, and a smaller $\beta$ a narrower one. To the left of the transition region, $y = 0$. To its right, $y = x$. Then, we enforce the gradients to be continuous by

$$(dy/dx)|_{x=-\beta} = 0, \quad (dy/dx)|_{x=\beta} = 1. \tag{8}$$

Applying the conditions gives the SmeLU activation

$$y_{\text{SmeLU}} = \begin{cases} 0; & x \leq -\beta \\ \frac{(x+\beta)^2}{4\beta}; & |x| \leq \beta \\ x; & x \geq \beta. \end{cases} \tag{9}$$

Fig. 1 (also Fig. 12) shows SmeLU and its continuous gradient as function of $\beta$. It matches ReLU on both sides, giving a gentle middle transition region, wider with greater $\beta$ and narrower with smaller $\beta$. SmeLU is a convolution of a ReLU with a box of magnitude $1/(2\beta)$ in $[-\beta, \beta]$. Its gradient is a hard Sigmoid. As with other smooth activations, we observe that with greater $\beta$, reproducibility is improved possibly at the expense of accuracy. Optimal accuracy on datasets we experimented with is for some $\beta > 0$. For $\beta \to 0$, SmeLU becomes ReLU. Fig. 2 demonstrates the surface of the output of a network with two inputs as function of the values of these inputs, with ReLU, SmeLU with different $\beta$, and no activation (top). Surfaces are obtained with weight normalization. The bottom row shows SmeLU with $\beta = 0.5$ for different $L_2$ weight norms. Examples of effects of normalization, clipping, and other activations on this surface are shown in Figs. 5-7 in Appendix A. Fig. 5 demonstrates the importance of normalization with clipping activations. Fig. 6 shows similar surfaces of other smooth activations. Fig. 7 shows effects of (other) normalizations. In Fig 2, we observe that ReLU exhibits multiple valleys (that lead to multiple optima in higher dimensions). Smoothing the activation by increasing $\beta$ smoothes these valleys. Thus with a smooth activation, there are fewer optima that the model can find while training, each spanning a larger region around it. However, activations that are too smooth resemble linear models with worse accuracy.

The parameter of SmeLU (as well as of other smooth activations) interacts with the normalization applied, as demonstrated in the second row of Fig. 2. Smoothing of the surface can be achieved by either tuning $\beta$ or the norm of the normalization applied. This, however, is only possible with smooth activations. Tuning weight norm with ReLU does not smoothen the surface. Thus the smooth activation allows for smoother surfaces whose smoothness (and good reproducibility behavior) can be tuned by either the parameters of the activation or by normalization. For consistency, one can keep the normalization to a norm of $v = 1$, and study the tradeoff as function of $\beta$, or apply SmeLU with no normalization (which may affect accuracy and reproducibility) and no clipping.

An approach that may improve the overall PD/accuracy tradeoff is to use wider values of $\beta$ for layers closer to the inputs and narrower $\beta$ values for layers closer to the output. Reproducibility seems to be dominated by rich parameter sets, and applying wider $\beta$ for layers that control such sets could improve it. Then, layers closer to the output can focus on better prediction accuracy. Alternatively, one can learn the values of $\beta$ for the whole model, per layer, or even for individual units, as part of the network training. Some insights on how this can be done are given in Appendix B.

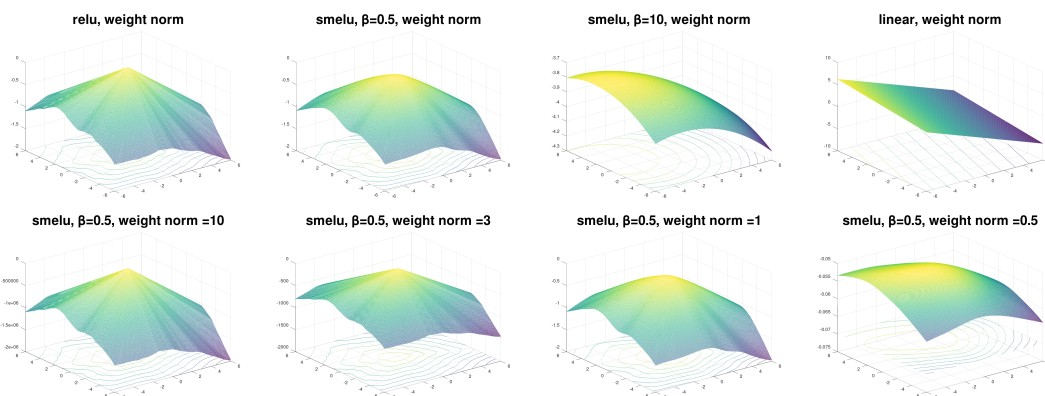

Figure 2: Three dimensional surfaces showing network outputs as function of 2-dimensional inputs in $[-6, 6]$, with 5 hidden layers of dimensions $[256, 128, 64, 32, 16]$ activated by ReLU, SmeLU with different $\beta$ parameters, and no activation (linear). Layers apply weight normalization, and there is no clipping of large activations. Matrices of all hidden layers are equal for all figures and are randomly drawn from a standard normal $\mathcal{N}(0, 1)$ distribution. Top: Different activations and SmeLU $\beta$ values. Bottom: SmeLU with $\beta = 0.5$ for different $L_2$ weight norms.

**Generalized SmeLU:** The constraints in (8) can be generalized to a wider family of activations, including ones such as a *leaky* SmeLU, by allowing different gradients $g_- \neq 0$ to the left and $g_+ > g_-$ to the right of the quadratic transition region, and allowing for an asymmetric transition region, and for a bias shift. The conditions are then

$$(dy/dx)|_{x=-\alpha} = g_-, \quad (dy/dx)|_{x=\beta} = g_+, \quad y(x = -\alpha) = t. \tag{10}$$

The five parameters $\{\alpha, \beta, g_-, g_+, t\}$ define a *generalized SmeLU*. Typically (but not necessarily), $\alpha, \beta > 0$, and $t \leq 0$ possibly to allow the activation to cross through the origin. For a leaky activation, $g_- > 0$, but we can also have a negative slope on the left (where then we relax the monotonicity requirement). Enforcing (10) and continuity of the function at $x = -\alpha$ and $x = \beta$, we come to the definition of the generalized SmeLU activation

$$y_{\text{gSmeLU}} = \begin{cases} g_- x + t + g_- \alpha; & x \leq -\alpha \\ ax^2 + bx + c; & -\alpha \leq x \leq \beta \\ g_+ x + t + \frac{\alpha+\beta}{2} g_- + \frac{\alpha-\beta}{2} g_+; & x \geq \beta \end{cases} \quad (11)$$

where

$$a = \frac{g_+ - g_-}{2(\alpha + \beta)}, \quad b = \frac{\alpha g_+ + \beta g_-}{\alpha + \beta}, \quad c = t + \frac{\alpha^2(g_+ + g_-) + 2\alpha\beta g_-}{2(\alpha + \beta)}. \quad (12)$$

The fifth pair of graphs in Fig. 1 (also Fig. 12) shows examples of generalized SmeLU with $t = 0$ and asymmetric transition regions, including leaky versions with $g_- > 0$ and also versions with $g_- \leq 0$. SmeLU is a special case of the generalized version, with $g_- = 0$, $g_+ = 1$, $\alpha = \beta$, and $t = 0$. Some additional special cases are reviewed in Appendix D.

The hyper-parameters of generalized SmeLU can be learned in training of a deep network together with the model parameters, either for the whole model, per layer, and even per neuron, as noted for SmeLU and discussed in Appendix B. Experiments demonstrate that for some datasets, layers closer to inputs may learn a negative $g_-$, implying that they learn some sparsification of the input, for better model accuracy. Layers closer to the output learned, in similar cases, slopes closer to $0$.

**Further Generalization - RESCU:** The general idea to derive the generalized SmeLU can be extended in various ways. Generally, we define *REctified Smooth Continuous Unit (RESCU)* as any function that consists of multiple smooth pieces, and is continuously differentiable. At any point, $\alpha$, $y(\alpha_-) = y(\alpha_+)$, as well as $(dy/dx)|_{\alpha_-} = (dy/dx)|_{\alpha_+}$. We can extend the generalized SmeLU by additional linear pieces and quadratic pieces connecting between any pair of linear pieces with different slopes. For example, the non-monotonicity of Swish (and alike) can be achieved by adding a concave piece to the left of the convex quadratic piece. At the point the gradient becomes $0$ it can be joined by a linear piece to its left with $0$ gradient.

Alternatively, the same concept can be used by joining pieces of more complex mathematical functions, ensuring smoothness at the connection points between the pieces. This makes activations like Swish special cases of RESCU with a single piece. SELU with $\beta = 1$ is another example with two pieces. Another example is the *REctified Sigmoid Continuous Unit*, which consists of a Sigmoid on the left, but linear on the right

$$y_{\text{Sig-RESCU}} = \begin{cases} 2\beta \cdot \sigma\left(\frac{2(x-\beta)}{\beta}\right); & x \leq \beta \\ x; & x \geq \beta \end{cases}. \quad (13)$$

We can apply the concept with polynomials of higher degree as well.

## 5 EXPERIMENTS

**Criteo:** We evaluated SmeLU and Swish on the benchmark Criteo Display Advertising Challenge dataset[1]. The dataset provides a binary classification task with $45M$ examples, which are split to $37M$ training examples, and validation and test sets of $4M$ examples each. Each example has 13 integer features and 26 categorical features. Models have 3 fully connected hidden layers of dimension 2572, 1454, 1596, respectively. Each categorical feature $x_k$ is hashed into $N_k$ buckets. If $N_k < 110$ the hash bin is encoded as a one-hot vector, otherwise it is embedded as a vector of dimension $d_k$. Values of $N_k$ and $d_k$ are taken from Ovadia et al. (2019) (see Appendix E). Each model is trained for one epoch using the Adam optimizer. Unlike Ovadia et al. (2019), input features are not fed into a batch-norm layer. Integer features are log-square-transformed. We trained models with ReLU, SmeLU and Swish activations. Each experiment consisted of 40 independent runs. For each run, models are initialized to different random values, and a different random shuffle is applied to the data. Training mini-batches of 1024 examples were used. Table 1 shows AUC and PDs; $\Delta_1$

---

[1]Criteo data in `https://www.kaggle.com/c/criteo-display-ad-challenge`

Table 1: Smooth activations on Criteo: AUC, and PDs; $\Delta_1$, $\Delta_1^r$.

| Model | | AUC | AUC stdev | $\Delta_1$ | $\Delta_1$ stdev | $\Delta_1^r$ % | $\Delta_1^r$ stdev % |
|---|---|---|---|---|---|---|---|
| ReLU | | 0.781 | 0.0144 | 0.053 | **0.033** | **36.3** | **19.2** |
| SmeLU | $\beta = 1$ | 0.783 | 0.0116 | 0.044 | 0.032 | 30.5 | 17.8 |
| | $\beta = 1.5$ | 0.785 | 0.0095 | 0.037 | 0.028 | 28.6 | 15.0 |
| | $\beta = 2$ | 0.786 | 0.0068 | 0.032 | 0.021 | 25.2 | 11.8 |
| | $\beta = 2.5$ | 0.787 | 0.0012 | **0.029** | **0.004** | **22.5** | **4.5** |
| | $\beta = 3$ | 0.787 | 0.0012 | **0.029** | 0.005 | 23.5 | 5.7 |
| | $\beta = 4$ | 0.787 | 0.0012 | **0.029** | 0.004 | 22.8 | 4.9 |
| Swish | $\beta = 0.25$ | 0.765 | 0.0208 | 0.088 | 0.006 | 45.1 | 29.4 |
| | $\beta = 0.5$ | 0.774 | 0.0197 | 0.078 | 0.025 | 45.3 | 25.1 |
| | $\beta = 1$ | 0.766 | 0.0211 | 0.087 | 0.009 | 45.3 | 26.5 |
| | $\beta = 2$ | 0.757 | 0.0208 | 0.078 | 0.028 | 40.0 | 29.1 |
| | $\beta = 4$ | 0.760 | 0.0170 | 0.076 | 0.022 | 39.5 | 27.1 |

(absolute) and $\Delta_1^r$ (relative), on the test data. ReLU exhibits huge PDs of over $36\%$. While only slightly improving AUC, around $40\%$ decrease in both PD measurements as well as substantial order of magnitude reductions in AUC and PD standard deviations, are observed with SmeLU. Tables 2 and 3 in Appendix E show that similar less extreme improvements are attained when training data is not shuffled, or models are identically initialized. Baseline ReLU PDs are lower but still over $20\%$. SmeLU still improves PD (about $12\%$ without shuffling) and AUC/PD standard deviations over ReLU. With Swish, on this dataset, we were unable to obtain comparable AUC and PD values.

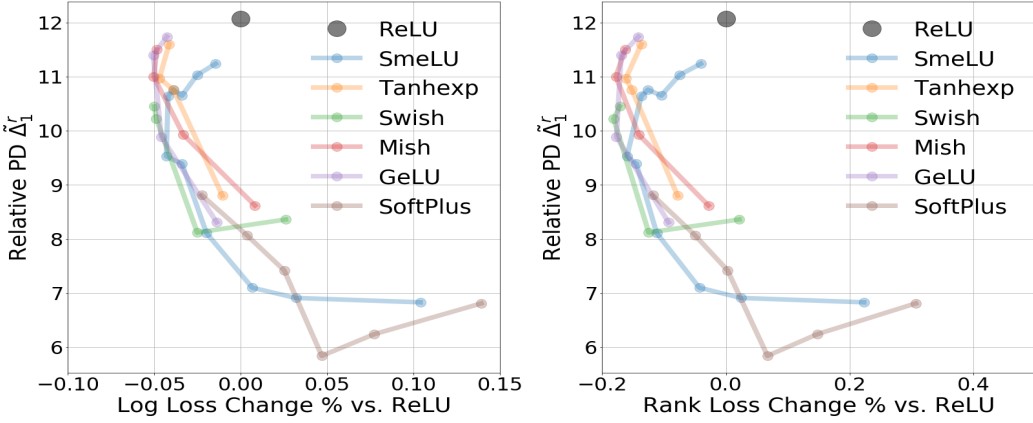

Figure 3: Relative PD; $\tilde{\Delta}_1^r$, expressed in [%] of the positive label prediction as function of loss change [%] from a ReLU baseline for different activations on real validation data. The X-axis is: left: logarithmic cross entorpy loss, right: ranking loss.

**Real Data:** We tested various smooth activations on a large private dataset for ad Click-Through-Rate (CTR) prediction for sponsored advertisement. We constructed a simple 6 layer fully connected network with dimensions $[1024, 512, 256, 128, 64, 16]$ with 5 informative features that are used as learned embedding vectors of different dimensions, providing a total of 240 inputs. Models are identically initialized, trained with data-shuffling on hundreds of billions of examples, using online mini-batch, single pass over the data, optimizing with the *Adagrad* optimizer (Duchi et al., 2011). Resources limit the quantity of models trained, so we use $M = 2$. Inference is applied to examples prior to training on them, and *progressive validation* (Blum et al., 1999), as commonly used for this problem (McMahan et al., 2013), is applied to measure validation logarithmic and ranking losses on predictions. Fig. 3 reports relative PD; $\tilde{\Delta}_1^r$, (as defined for binary labels) as function of both average validation logarithmic (cross-entropy) loss (left) and ranking loss (right). Losses are expressed as percentage of the baseline ReLU model. Relative PD is expressed as percentage of the actual positive label prediction. On the graph, moving left implies better accuracy, and moving down

implies better reproducibility. For smooth activations, the different points are the result of different $\beta$. Moving down at least before reaching an optimum implies larger $\beta$ for SmeLU and smaller $\beta$ for the other activations. In such a large scale system, we observe major relative PD values of over 10% with ReLU (12% when models initialized equally, and 13% when initialized differently). Smooth activations are substantially better on PD and better on the tradeoffs than ReLU. Among themselves, the tradeoffs are more comparable, where SmeLU, GELU and Swish appear to be slightly better on this data than Mish and Tanhexp. SoftPlus appears to push towards better PD, but at the expense of accuracy with unacceptable trade-offs (in such a system even a fraction of percent can make a difference). Continuously differential SELU appears inferior in accuracy and PD. SmeLU also seems to be stronger on PD, whereas Swish on accuracy. Results shown use weight normalization with norm $v = 1$ without clipping. We observe similar relative behavior without normalization (no clipping). However, PD's of smooth activations increase, but are still better than the ReLU baseline.

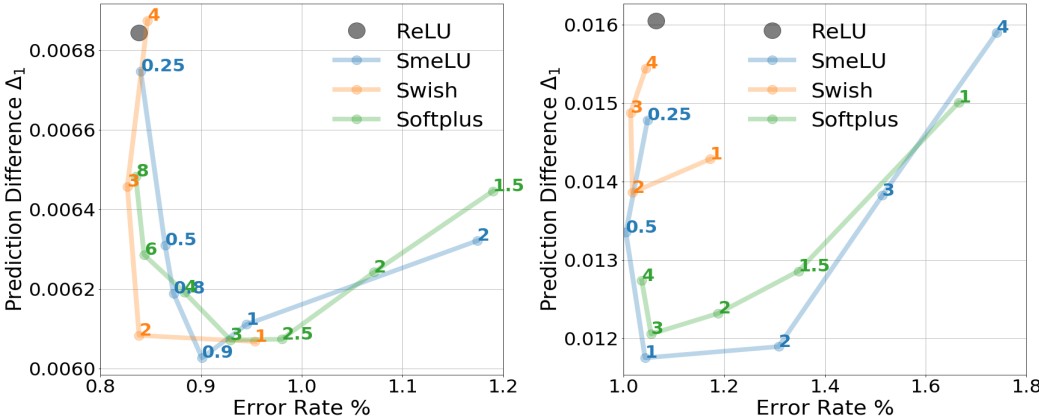

Figure 4: PD $\Delta_1$ as function of error rate on MNIST dataset for different activations with different $\beta$ parameters. Left: with Adagrad optimizer, right: with an SGD optimizer.

**Benchmark MNIST Dataset:** We tested PD and accuracy on the MNIST dataset (LeCun, 1998). Models were implemented with Tensorflow Keras. Networks were simple 2 layer networks, with layers of width 1200. We regularized with dropout (input 0.2, layers 0.5) and image augmentation in training (starting from epoch 2, two dimensional independent random shifts by $\{-3, -2, -1, 0, 1, 2, 3\}$ pixels take place with probability 0.5.) Hidden weights were identically initialized with the default Keras uniform random, and biases to 0. For AdaGrad, we used learning rate 0.02, accumulator initialization 0.1, and 150 epochs without weight normalization. For SGD, we used learning rate 0.01, momentum 0.9, and 50 epochs. Batch size is 32. For each experiment, we trained $M = 12$ models. Fig. 4 shows PD $\Delta_1$ as function of error rate for both configurations, AdaGrad (left) and SGD (right). Graphs for other PD metrics, with similar behavior, are in Appendix F. PD values may not be as large as they are in large scale data, but we still see PD change substantially relative to its lowest values (over 10% for AdaGrad, over 30% for SGD). As in large scale, ReLU's PD is much worse than those achievable by smooth activations, and among themselves, smooth activations are comparable. With SGD, however, Swish appears inferior on PD.

## 6 Conclusions

We studied irreproducibility in deep networks, showing that the popular ReLU activation is a major factor in exacerbating prediction differences. Smooth activations allow substantial improvement, giving better accuracy-reproducibility tradeoffs. We introduced SmeLU, and its generalizations, specifically demonstrating that superior accuracy-reproducibility tradeoffs are achievable even with very simple inexpensive to implement activations, whose performance on both accuracy and PD do not fall from and can be better than those of more complex expensive to implement, and non-monotonic, smooth activations. We empirically demonstrated these tradeoffs, showing that proper smooth activations are superior to ReLU. For a specific dataset, model architecture, and model configuration, one smooth activation may be better than the other. We have shown, however, that the much simpler SmeLU can be comparable or superior to the more complex activations.

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

## A    SMOOTH ACTIVATIONS AND NORMALIZATION INDUCED SURFACES

In this appendix, we demonstrate effects of smooth activations and normalization on the objective surfaces, enhancing on the description in Sections 3-4, and on the illustrations of Fig. 2. Fig. 5 expands on Fig. 2, showing objective surfaces for different activations, parameters and normalizations. It shows the transition from a non-smooth ReLU to a linear activation through different parameters of SmeLU. As we increase $\beta$, the objective surface becomes smoother. It also demonstrates the importance of normalization, especially when activations are clipped to some cap. The first row duplicates that of Fig. 2 for completeness. The second row shows the transition of a weight normalized clipped activation from ReLU to linear through widening SmeLUs. With weight normalization there are slight deviations from the top row, but the behavior appears rather similar. The third row shows the same transition without weight normalization and clipping. In order to be as smooth, a larger $\beta$ must be used than in the first row. This result is in line with the experiments reported in Section 5 on real data, where PD values increased in all cases with the same $\beta$ parameters if weight normalization was not applied. Furthermore, the objective variance is now much larger than in the first row. When clipping is added, as shown in the last row, the effect of the smoothness disappears, where numerous peaks appear in all activations. With large $\beta$, the effectiveness of the activation altogether also appears to diminish. This happens even with the linear activation. As observed in the second row, weight normalization diminishes the effect of clipping.

Fig. 6 shows surfaces for Swish, SoftPlus, GELU and SELU with weight normalization, without clipping. For this simple two dimensional input example, the smooth activations appear rather similar to the surfaces of SmeLU. As mentioned earlier, the smoothness is reversed as a function of $\beta$ from SmeLU, with the larger $\beta$ values being less smooth. Since GELU can be approximated by Swish with a larger parameter, for the same $\beta$, Swish is smoother. Very similar surfaces are observed for Mish and Tanhexp. SELU is smooth for $\beta = 1$, but is clearly not smooth for other values of $\beta$. This is observed in Fig. 6 for $\beta = 10$.

Fig. 7 demonstrates the effect of normalization (weight and layer) on no activation and on ReLU, and the effects of layer normalization (Ba et al., 2016) on SmeLU. Unlike weight normalization, layer normalization changes the shape of the surface from the shape obtained when there is no normalization. While the type of norm changes the shape of the surface obtained, unlike smooth activations, the norm value $v$ has no effect on the shape of the surface observed for either no activation or ReLU. (It does affect the magnitude of the objective, though.) For smooth activations, on the other hand, as observed in Fig. 2, the magnitude $v$ of the norm does affect the smoothness of the surface also with layer normalization, as demonstrated for SmeLU on the bottom row, with smoother objective as the norm is smaller.

## B    LEARNING SMELU PARAMETERS

As we observed, $\beta$ in smooth activations roughly serves as a tuning knob between accuracy and reproducibility. However, the optimal accuracy is attained away from the end point (0 for SmeLU,

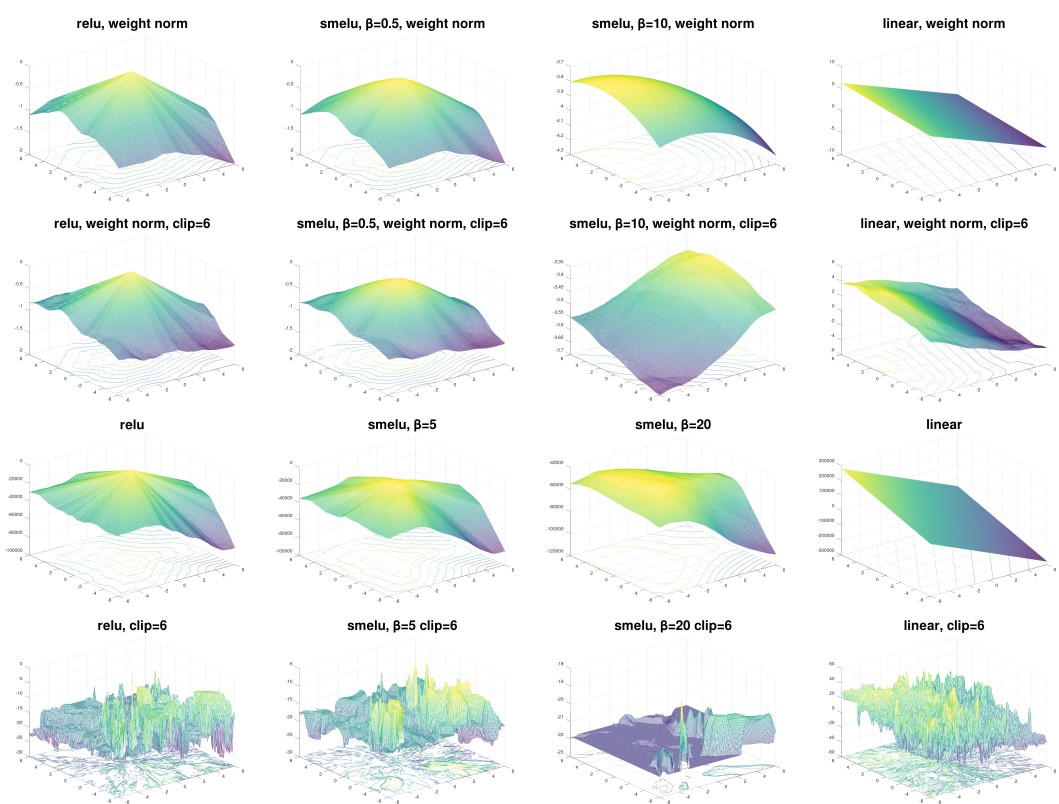

Figure 5: Three dimensional surfaces showing network outputs as function of 2-dimensional inputs in $[-6, 6]$, with 5 hidden layers of dimensions $[256, 128, 64, 32, 16]$ with no activation (linear), and activated by ReLU, and SmeLU with different $\beta$ parameters. Matrices of all hidden layers are equal for all figures and are randomly drawn from a standard normal $\mathcal{N}(0, 1)$ distribution. Top: ReLU, SmeLU different $\beta$ and linear, no clipping of pre-activation values with $L_2$ norm 1 weight normalization. Second Row: weight normalization with activation clipping to $[-6, 6]$. Third Row: no clipping, no weight normalization. Bottom Row: clipping, no weight normalization.

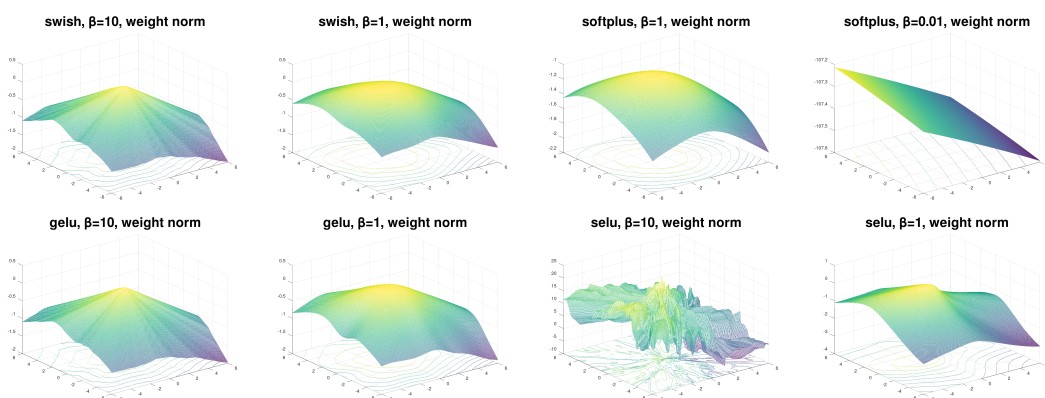

Figure 6: Three dimensional surfaces with identical configuration to Fig. 5 for Swish, Softplus, GELU, and SELU, with different $\beta$ parameters, all with weight normalization and no clipping. Top: Swish and Softplus. Bottom: GELU and SELU.

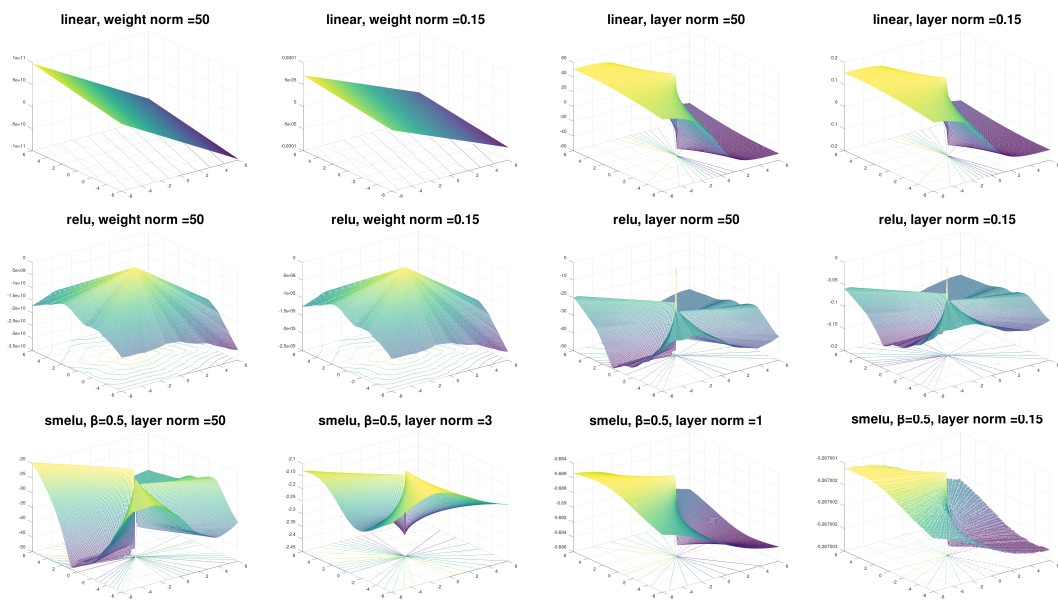

Figure 7: Three dimensional surfaces with identical configuration to Fig. 5 for no activation, ReLU and Smelu ($\beta = 0.5$) with weight and layer normalizations, with no clipping. Top: Linear with different weight and layer norms. Middle: ReLU with different weight and layer norms. Bottom: SmeLU with different layer norms.

$\infty$ for Swish and other smooth activations). Similarly, there are no guarantees that the wider is the activation the more reproducible it is, at the other extreme. It is expected that the optimal points will be properties of both the actual dataset and the model architecture. For the generalized SmeLU and wider RESCU generalization, more parameters should be tuned. While tuning can optimize various tasks, we focus here on tuning for accuracy and reproducibility.

The parameter $\beta$ or the parameters of the generalized SmeLU $(\alpha, \beta, g_-, g_+, t)$ can be trained and learned as part of the training process of a deep network. Learning can optimize accuracy, reproducibility or any other objective. Training can learn a single parameter set for the whole network, a single parameter set per layer, or a single parameter set per neuron. Depending on the task and number of updates, learning rates can be tuned for each of the learned hyper-parameters. Initialization can initialize to known good parameters. For the generalized SmeLU, we can initialize values to those of a reasonable SmeLU, e.g., with $g_- = 0$, $g_+ = 1$, $t = 0$, and $\alpha = \beta = 0.5$. Empirically, we observe, when optimizing for accuracy, that each layer learns a different set of parameters. Specifically, we observed that layers closer to the input tend to learn a negative $g_-$. The first hidden layer that is connected to the input (especially in systems where the input consists of learned embeddings) feeds input that is usually more symmetric around the origin, while layers farther from the input see biased inputs. With the learned $g_-$, it appears that the layer closest to the input attempts to learn sparsification, forcing its inputs either to an extreme or to the origin.

Learning hyper-parameters for accuracy can be done simultaneously with training the parameters of the networks. Alternatively, this can be done in one training session. Then, the hyper-parameters can be fixed, and used for training the parameters. The procedure can be further applied to learning functional forms of the activation as done by Ramachandran et al. (2017). One can apply such a methodology on a RESCU activation, where the pieces are learned under the constraints of continuous function and gradient between the pieces.

Optimization of the parameters can combine an objective for both accuracy and reproducibility, as a mean or weighted sum of both objectives. However, this must be done carefully, as these objectives may be conflicting. Optimizing the activation for reproducibility may be more involved, as gradient methods with the objective loss used for training neural networks tend to try to improve the model's accuracy. There are several options for training the activation parameters to improve reproducibility. We can include reproducibility optimization of the parameters in training the full network, or train

the model offline while learning the parameters that are better for reproducibility first, and then repeat training to learn the model itself using these parameters. In either case, *ensembles* can be used as a proxy for reproducibility, where we impose a loss that minimizes the prediction difference (either in probability or on log-odds score) between components of the ensemble. This approach essentially applies *co-distillation* (Anil et al., 2018) but for the purpose of learning the hyper-parameters that improve reproducibility instead of training the actual model. Co-trained ensemble may not mimic the full effect causing divergence of models as they may have less randomness in the order of examples they see and updates they apply, but empirical results show that even with that components of ensemble tend to diverge from one another, even when initialized identically, and especially when initialized differently. Unlike co-distillation, to learn the hyper-parameters for better reproducibility, the objective loss in training the ensemble is applied only on the parameters, but the similarity loss is applied only on the hyper-parameters.

There are disadvantages to learning the hyper-parameters while training. This is because of potentially conflicting objectives to accuracy and reproducibility. Furthermore, in many cases, the preferred deployed model should not be an ensemble. We can use an ensemble to optimize the hyper-parameters for reproducibility offline, and then train and deploy a model that retrains with the reproducibility optimized activation hyper-parameters.

If we deploy an ensemble, training with identically initialized components may be valuable for optimizing activation hyper-parameters for better reproducibility, but will be suboptimal for deployment, at least for better reproducibility. As shown by Shamir & Coviello (2020), ensembles are advantageous for reproducibility by leveraging and encouraging the differences between their components, so that as a whole, the ensemble captures the objective space better. Initializing components identically for optimizing the hyper-parameters defeats this idea, and thus it is more desirable to apply such initialization only when learning the activation hyper-parameters, but then retrain the models with these parameters with different initializations of the components, for better deployed reproducibility.

## C  GRAPHS OF DIFFERENT SMOOTH ACTIVATIONS

In this appendix, we provide activation function and gradient curves for the different (smooth) activations discussed in this paper as function of their parameter. This set of curves completes the ones shown in Fig. 1. Fig 8 shows curves for the different activations with $\beta = 1$. Figs. 9, 10, 11 and 12 show SELU and CELU, SoftPlus, Swish, GELU, Mish, Tanhexp, SmeLU, generalized SmeLU, and Sigmoid-RESCU, respectively. SELU is shown to be smooth only for $\beta = 1$. The equations for CELU are given by

$$y_{\text{CELU}} = \begin{cases} x, & \text{if } x \geq 0 \\ \beta \cdot \left( \exp\left( \frac{x}{\beta} \right) - 1 \right), & \text{if } x < 0. \end{cases} \tag{14}$$

We observe the close similarity between Swish, GELU, Mish, and Tanhexp, and the relation between Sigmoid-RESCU and SmeLU. Subsequently to our work, and inspired by SmeLU, (Lin & Shamir, 2019), a different SmoothRelu activation was given in Xie et al. (2020),

$$y_{\text{SmoothRelu}} = \begin{cases} x - \frac{1}{\beta} \log \left( \beta x + 1 \right), & \text{if } x \geq 0, \\ 0, & \text{if } x < 0. \end{cases} \tag{15}$$

This activation is also a RESCU activation. Graphs for this activation are included in Fig. 12. As observed, this activation does not achieve full slope 1, unless $\beta \gg 1$, in which case, it already approaches ReLU. Thus it may lack the smoothness tuning flexibilities of SmeLU.

## D  GENERALIZED SMELU - REFORMULATIONS AND SOME SPECIAL CASES

In this appendix we show some special cases of the generalized SmeLU defined in (11), and also address the asymptotic relation between SmeLU and SoftPlus.

### D.1  ASYMMETRIC SMELU

SmeLU is a special case of the generalized SmeLU. The definition of the generalized SmeLU in (11) can be used to obtain a version of SmeLU that is asymmetric around the origin. Like SmeLU,

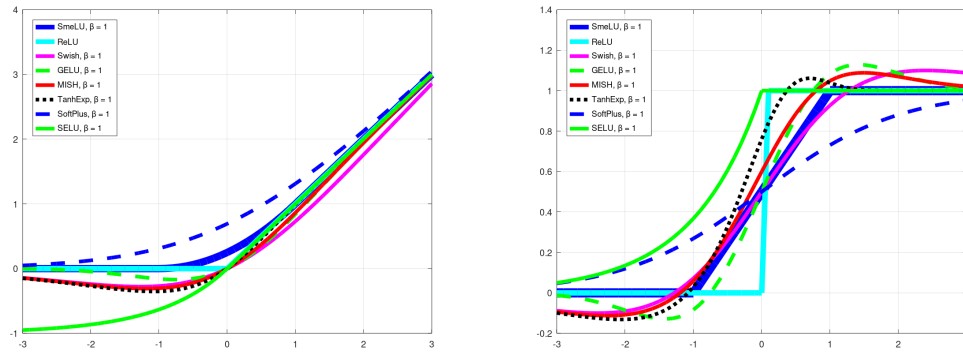

Figure 8: Different activations and their gradients for $\beta = 1$.

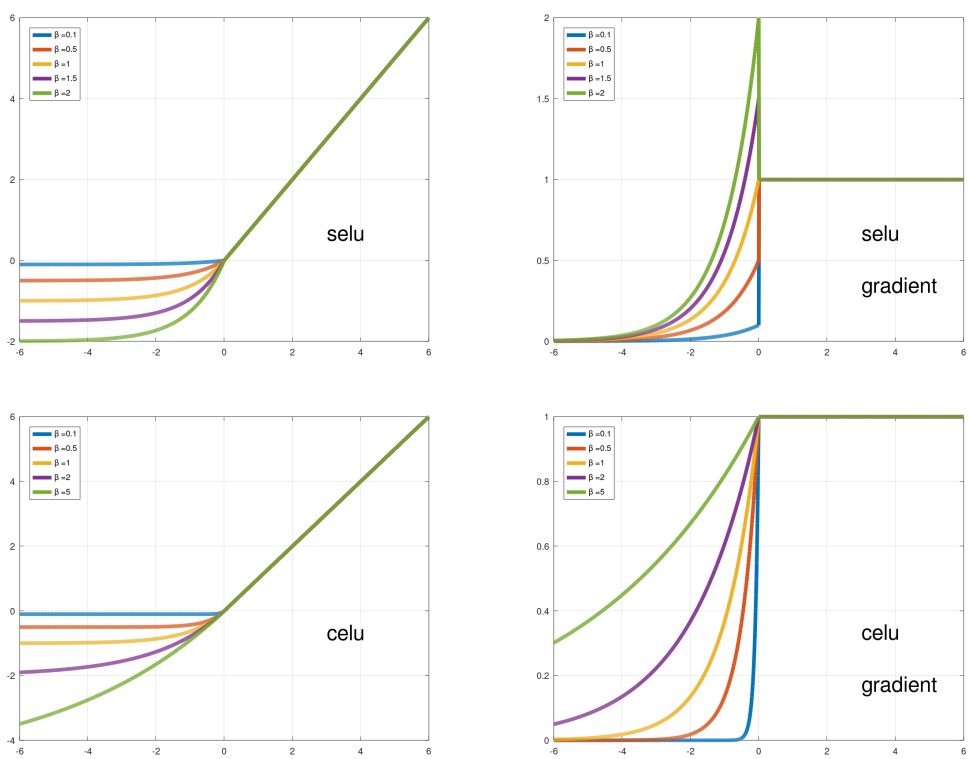

Figure 9: SELU (top) and CELU (bottom) activations and gradients with different $\beta$ parameter values.

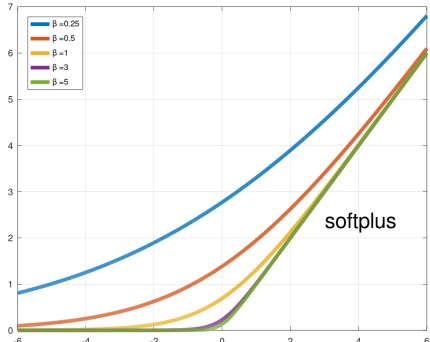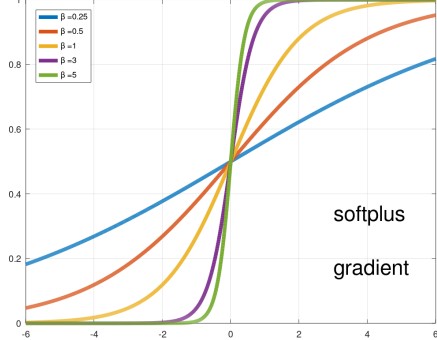

Figure 10: SoftPlus activations and gradients with different $\beta$ parameter values.

this activation can guarantee $0$ output on the left, and slope $1$ on the right, but a transition region in $[-\alpha, \beta]$, where $\alpha \neq \beta$,

$$
y_{\text{Asymmetric-SmeLU}} = \begin{cases} 0; & x \leq -\alpha \\ \frac{(x+\alpha)^2}{2(\alpha+\beta)}; & -\alpha \leq x \leq \beta \\ x + \frac{\alpha-\beta}{2}; & x \geq \beta. \end{cases} \tag{16}
$$

Setting $\alpha = \beta$ gives SmeLU.

### D.2 LEAKY SMELU

Setting $g_- > 0$, but $g_+ = 1 > g_-$, $\alpha = \beta$ and $t = 0$, gives the SmeLU version of a *Leaky ReLU*, referred to as *Leaky SmeLU*, and given by

$$
y_{\text{Leaky-SmeLU}} = \begin{cases} g_- \cdot (x + \beta); & x \leq -\beta \\ \frac{1-g_-}{4\beta} \cdot x^2 + \frac{1+g_-}{2} \cdot x + \frac{\beta}{4}(1+3g_-); & |x| \leq \beta \\ x + g_- \cdot \beta; & x \geq \beta. \end{cases} \tag{17}
$$

Apart for the middle region, this activation resembles Leaky ReLU.

### D.3 SHIFTED SMELU

The constraint used to generate the generalized SmeLU constrains the $g_-$ slope region to end at the point $x = -\alpha$, $y = t$. Constraints can be specified differently, where this region ends at the origin, and then shifted horizontally and/or vertically. Vertical shift is given by $t$, whereas we can define the activation $z(x) = y(x - s)$ for a horizontal shift by $s$.

### D.4 ORIGIN CROSSING SMELU

SmeLU as defined in (9) does not cross the origin. We can similarly define a version of generalized SmeLU that is constrained to cross the origin instead of a point $(-\alpha, t)$. We can apply the generalized SmeLU with the desired $\alpha$, $\beta$ and gradients, and then shift it vertically to cross the origin (or we can solve directly with the new constraints). A zero crossing version keeps the sign of the input on one hand, but does not suppress negative values to $0$. The form is given by

$$
y_{\text{zero-cross-SmeLU}} = \begin{cases} g_- \cdot x + \frac{\alpha^2(g_- - g_+)}{2(\alpha+\beta)}; & x \leq -\alpha \\ \frac{g_+ - g_-}{2(\alpha+\beta)} x^2 + \frac{\alpha g_+ + \beta g_-}{\alpha+\beta} x; & -\alpha \leq x \leq \beta \\ g_+ \cdot x + \frac{\beta^2(g_- - g_+)}{2(\alpha+\beta)}; & x \geq \beta. \end{cases} \tag{18}
$$

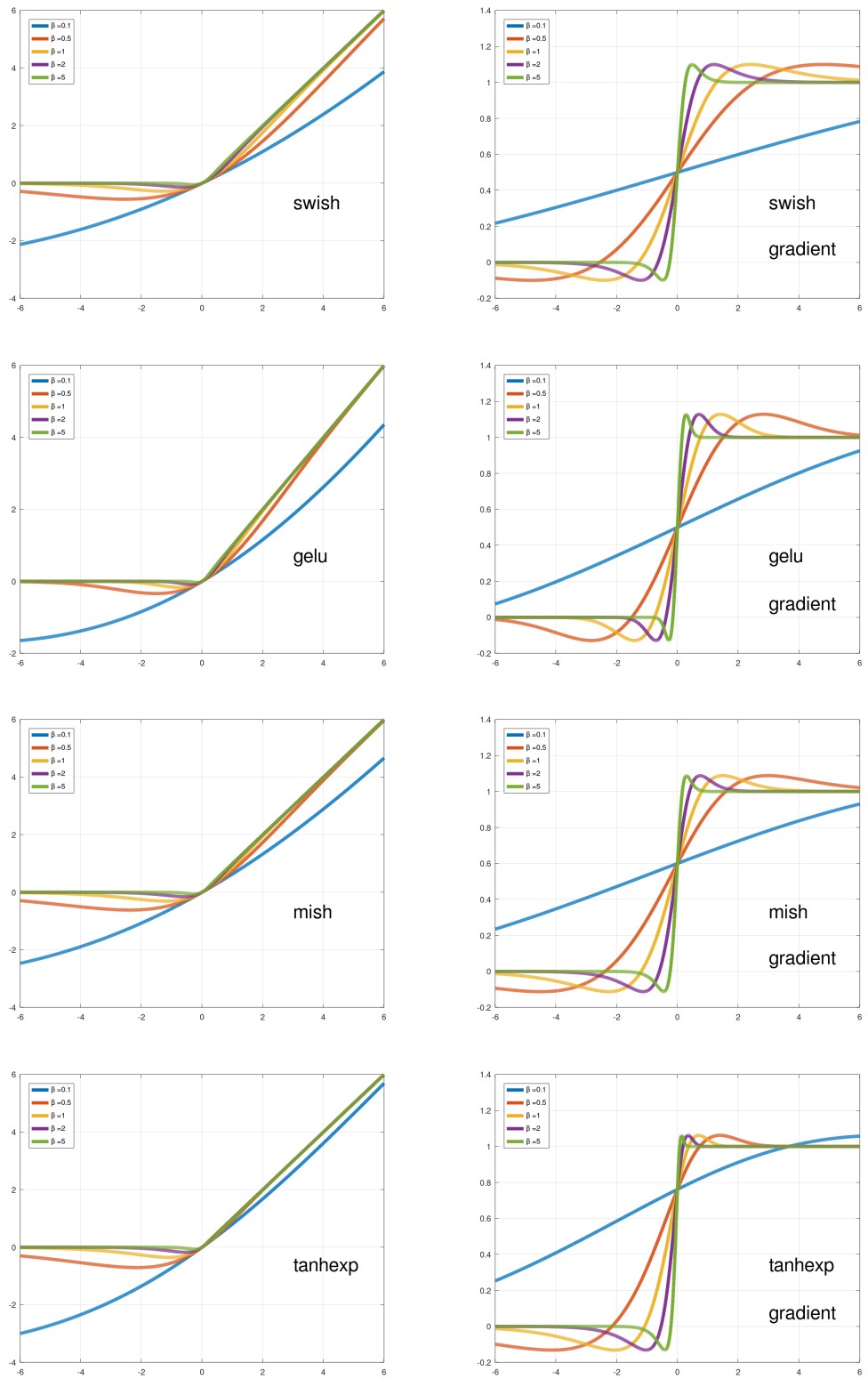

Figure 11: Swish, GELU, Mish, and TanhExp activations and gradients with different $\beta$ parameter values.

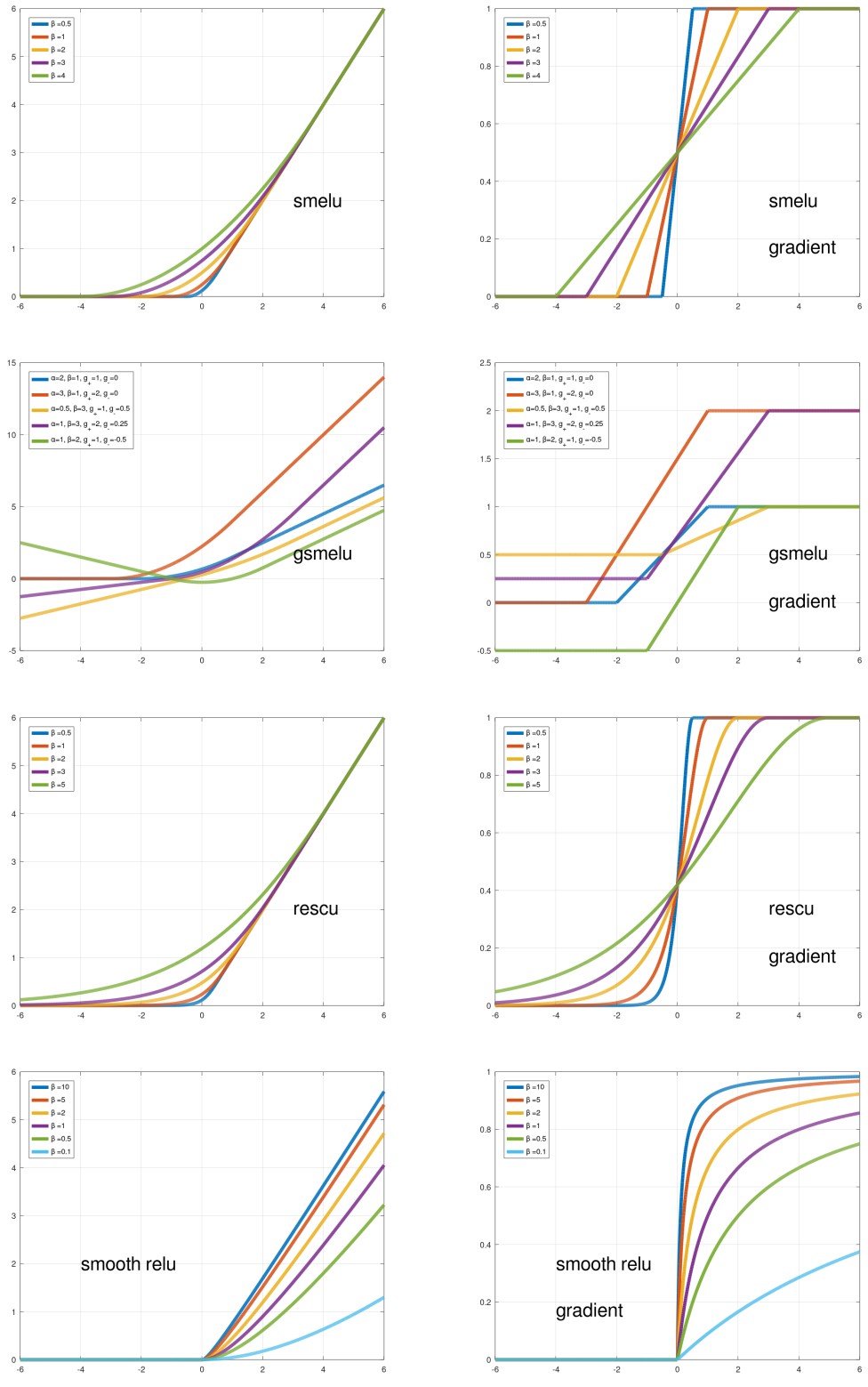

Figure 12: SmeLU, Generalized SmeLU, Sigmoid-RESCU, and SmoothRelu activations and gradients with different $\alpha$, $\beta$ and gradient parameter values.

### D.5 Asymptotic Relation of SmeLU and SoftPlus

The gradient of the SmeLU function is a hard Sigmoid. This already shows a connection to SoftPlus, whose gradient is a Sigmoid. To further demonstrate the relation between SmeLU and SoftPlus, we express SoftPlus using a reciprocal parameter $\gamma$ to the original definition in (3),

$$y_{\text{SoftPlus}} = \gamma \ln\left(1 + \exp(x/\gamma)\right). \tag{19}$$

Asymptotically, for $x \to -\infty$, we have $y \to 0$. For $x \gg 0$, $y \approx x$. For $|x| \ll 1$, we can approximate the SoftPlus using Taylor series expansion by

$$y_{\text{SoftPlus}} \approx \gamma \ln 2 + \frac{x}{2} + \frac{x^2}{8\gamma}. \tag{20}$$

Assigning the SmeLU parameter $\beta = 2\gamma$, we obtain for $|x| \ll 1$,

$$y_{\text{SoftPlus}}(x) - y_{\text{SmeLU}}(x) \approx \gamma \ln 2 - \frac{\beta}{4} = \frac{\beta}{2}\left(\ln 2 - \frac{1}{2}\right) = \frac{\beta}{4} \ln \frac{4}{e}. \tag{21}$$

Thus the middle region around $x = 0$ of SoftPlus is a vertical shift by $\beta/4 \ln(4/e)$ of the middle region of SmeLU with a factor of 2 wider parameter, but the extremes on both sides meet those of SmeLU. Therefore, SoftPlus opens wider, away from the origin, making it even smoother for the same parameter, on one hand, but also potentially less accurate on the other, due to the distance of the middle region from the origin.

## E   Experiments on the Criteo Dataset

We describe the set up of the experiments on the Criteo benchmark dataset and provide additional results.

**Model architecture:** Models have 3 fully connected hidden layers of dimension 2572, 1454, 1596 respectively. Each categorical feature $x_k$ is hashed into $N_k$ buckets. If $N_k < 110$ the hash bin is encoded as a one-hot vector, otherwise it is embedded as a vector of dimension $d_k$. Values of $N_k$ and $d_k$ are taken from Ovadia et al. (2019) and are the following (where $d_k = 0$ means that the hash bin is encoded as a one-hot vector):

$$
\begin{aligned}
Nk \;=\; & [1373, 2148, 4847, 9781, 396, 28, 3591, 2798, 14, 7403, 2511, 5598, 9501, \\
& 46, 4753, 4056, 23, 3828, 5856, 12, 4226, 23, 61, 3098, 494, 5087] \qquad\qquad (22) \\
dk \;=\; & [3, 9, 29, 11, 17, 0, 14, 4, 0, 12, 19, 24, 29, 0, 13, 25, 0, 8, 29, 0, 22, 0, 0, 31, 0, 29] \;\;(23)
\end{aligned}
$$

Unlike the set up in Ovadia et al. (2019), input features are not fed into a batch-norm layer, while integer features are log-square-transformed.

**Additional results:** Table 2 shows results for ReLU, SmeLU and Swish when the training data is not shuffled. As a consequence, all models visit the data in the same order of batches, but are initialized differently. (Note, however, that due to randomness in the platform, within a batch, there could still be some randomness in the order examples are being processed.) Table 3 shows results for ReLU, SmeLU and Swish when all models have the same initialization of the hidden layer weights. In both cases, models exhibit irreproducibility. Without shuffling, as a result of initialization (and random effect in the platform that applies the updates). With identical initialization, irreproducibility is due to the training data shuffling. In both case, SmeLU improves $\Delta_1$ and $\Delta_1^r$ by about 12%, and exhibits a factor of 4 reduction of the standard deviations of both AUC and $\Delta_1$.

## F   Additional results on MNIST

In this appendix, we show matching graphs to those in Fig. 4 for the other PD metrics; $\Delta_2$, $\Delta_1^L$, and $\Delta^H$, in Figs 13, 14, and 15, respectively. The behavior of the different activations shown in these figures for all the metrics is very similar to the behavior w.r.t. $\Delta_1$.

Table 2: Smooth activations on Criteo: AUC, $\Delta_1$, $\Delta_1^r$. Training data is not shuffled.

| Model | | AUC | AUC stdev | $\Delta_1$ | $\Delta_1$ stdev | $\Delta_1^r$ % | $\Delta_1^r$ stdev |
|---|---|---|---|---|---|---|---|
| ReLU | | 0.786 | 0.0068 | 0.028 | 0.014 | **21.5** | **8.3** |
| SmeLU | $\beta = 1$ | 0.787 | 0.0008 | **0.025** | 0.003 | 19.3 | 3.4 |
| | $\beta = 1.5$ | 0.787 | 0.0007 | **0.025** | 0.003 | 19.0 | 2.9 |
| | $\beta = 2$ | 0.787 | 0.0011 | **0.025** | 0.004 | 19.1 | 3.7 |
| | $\beta = 2.5$ | 0.786 | 0.0011 | **0.025** | 0.003 | 19.2 | 2.8 |
| | $\beta = 3$ | 0.787 | 0.0011 | **0.025** | 0.003 | **18.4** | **2.8** |
| | $\beta = 4$ | 0.787 | 0.0007 | **0.025** | 0.003 | 20.5 | 4.6 |
| Swish | $\beta = 0.25$ | 0.769 | 0.0203 | 0.087 | 0.018 | 47.2 | 28.3 |
| | $\beta = 0.5$ | 0.765 | 0.0216 | 0.087 | 0.006 | 44.3 | 32.8 |
| | $\beta = 1$ | 0.768 | 0.0215 | 0.085 | 0.012 | 46.2 | 30.1 |
| | $\beta = 2$ | 0.756 | 0.0180 | 0.070 | 0.030 | 37.4 | 29.9 |
| | $\beta = 4$ | 0.764 | 0.0180 | 0.082 | 0.010 | 46.6 | 23.1 |

Table 3: Smooth activations on Criteo: AUC, $\Delta_1$, $\Delta_1^r$. Models are identically initialized.

| Model | | AUC | AUC stdev | $\Delta_1$ | $\Delta_1$ stdev | $\Delta_1^r$ % | $\Delta_1^r$ stdev |
|---|---|---|---|---|---|---|---|
| ReLU | | 0.787 | 0.0023 | 0.031 | 0.007 | **25.3** | 7.2 |
| SmeLU | $\beta = 1$ | 0.787 | 0.0004 | 0.028 | 0.004 | 22.2 | **5.0** |
| | $\beta = 1.5$ | 0.787 | 0.0004 | 0.029 | 0.004 | 23.5 | 5.8 |
| | $\beta = 2$ | 0.787 | 0.0003 | 0.029 | 0.004 | 23.6 | 6.4 |
| | $\beta = 2.5$ | 0.787 | 0.0005 | 0.028 | 0.004 | 22.5 | 5.6 |
| | $\beta = 3$ | 0.787 | 0.0004 | **0.027** | 0.004 | **22.4** | 5.6 |
| | $\beta = 4$ | 0.787 | 0.0003 | **0.027** | 0.004 | **22.4** | 5.7 |
| Swish | $\beta = 0.25$ | 0.767 | 0.0210 | 0.088 | 0.012 | 47.5 | 28.3 |
| | $\beta = 0.5$ | 0.770 | 0.0207 | 0.083 | 0.015 | 45.9 | 27.5 |
| | $\beta = 1$ | 0.769 | 0.0204 | 0.084 | 0.013 | 46.7 | 26.1 |
| | $\beta = 2$ | 0.763 | 0.0204 | 0.085 | 0.012 | 43.4 | 27.9 |
| | $\beta = 4$ | 0.762 | 0.0181 | 0.083 | 0.014 | 45.3 | 24.3 |

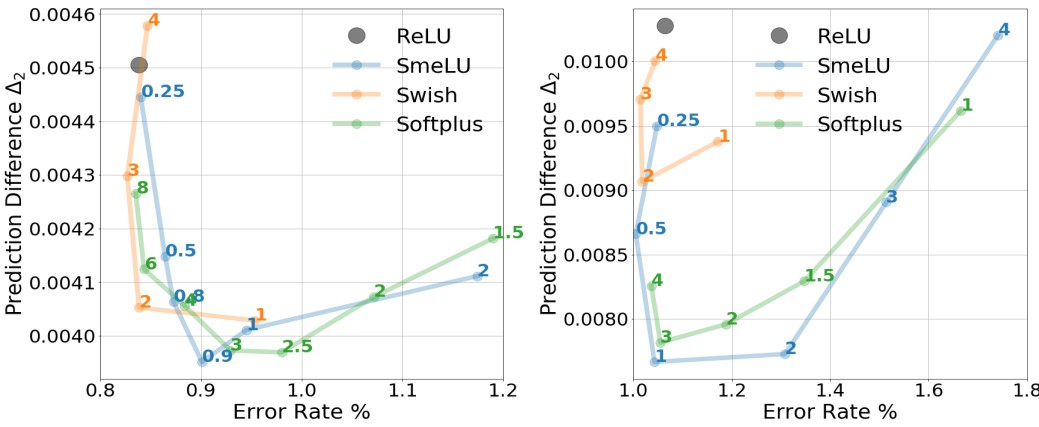

Figure 13: PD $\Delta_2$ as function of error rate on MNIST dataset for different activations with different $\beta$ parameters. Left: with Adagrad optimizer, right: with an SGD optimizer.

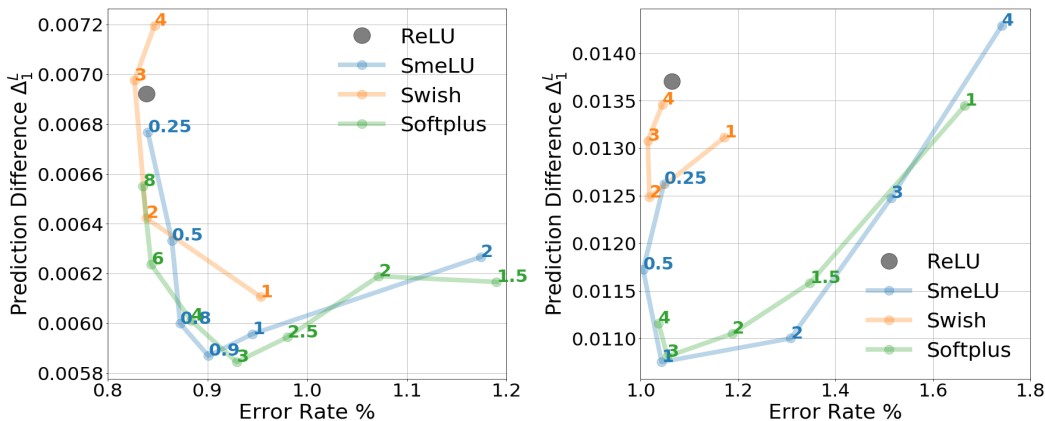

Figure 14: PD $\Delta_1^L$ as function of error rate on MNIST dataset for different activations with different $\beta$ parameters. Left: with Adagrad optimizer, right: with an SGD optimizer.

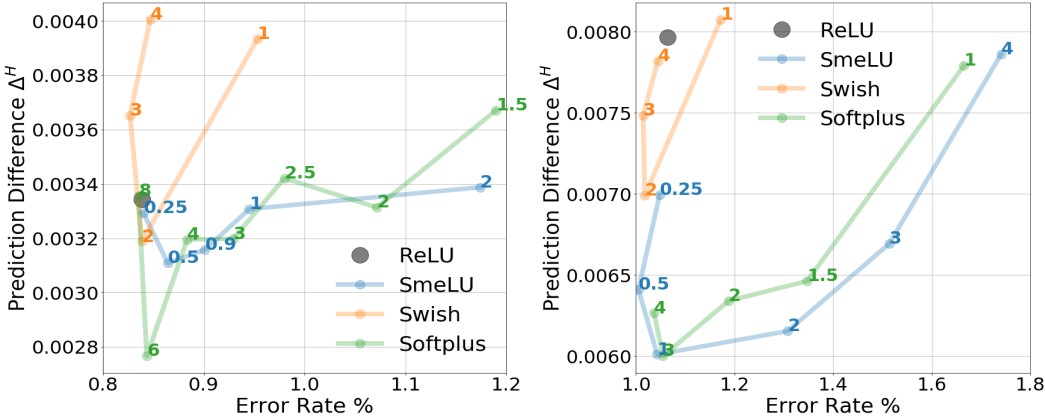

Figure 15: PD $\Delta^H$ as function of error rate on MNIST dataset for different activations with different $\beta$ parameters. Left: with Adagrad optimizer, right: with an SGD optimizer.

