# OpenReview forum: "Smooth Activations and Reproducibility in Deep Networks"
_ICLR.cc/2021/Conference — Reject_

### Official Review · AnonReviewer3 · 2020-10-27
**The idea that smooth activation functions may have better reproducibility of neural networks is interesting. However, the experiments need more elaborations.**

**Rating:** 4
**Confidence:** 3

**Review:**

Thanks for the efforts of the authors. Some of my concerns have been addressed. However, I still think that the experiments are not convincing enough for ICLR. So I keep my score.
_____________________________________________
Summary:
The paper argues that a smooth activation function may produce a smooth surface of the output of a network, which identifies a good reproducibility behavior. Based on this observation, the paper proposes the SmeLU activation function and its generalized version. Experiments with fully-connected neural networks are presented.

Strengths:
+ The idea that smooth activation functions may have better reproducibility of neural networks is very interesting. It provides a different way to understand the role of activation functions in neural networks.

Weaknesses:
- Figure 2 shows the motivation of the paper. It uses the surface of the output of a neural network w.r.t. its input to show the number of local minima. However, we usually use loss landscape to show the local minima of a neural network. Moreover, the paper needs to test with various layers (e.g., shallow and deep), initializations (e.g., Gaussian or Uniform), and architectures. It seems that Figure 2 only works with networks having two-dimension input and one-dimension output. For other networks, the visualization of the loss landscape of neural networks may be a good reference.
- For SmeLU, the parameter \beta is very important. It balances the accuracy and the reproducibility. However, the paper does not give a practical method to choose its value. The paper claims in Appendix B that \beta can be learned with the weights of a neural network. However, the commonly used objective functions correspond to the accuracy. How to update \beta to balance the accuracy and the reproducibility during training is unknown.
-In Figure 4, different optimizers could produce different results. However, the paper only uses Adagrad. It is better to test with SGD, Adagrad, Adam, and AMSGrad.
-The datasets used in the paper is very small. It is better to test with larger datasets.
-The paper only tests with fully-connected neural networks. It is better to test with convolutional neural networks.

Overall, since the paper is not a theoretical one, it needs extensive experiments to verify the claims. However, the experiments in the paper are not convincing enough.

---

> ### Author Response · Authors · 2020-11-24
> **Response to AnonReviewer3**
>
> We thank the reviewer for his efforts, time and comments, which we address below. We included more empirical results in the revised version, with the Criteo benchmark dataset, that clearly demonstrate irreproducibility and the positive effects of SmeLU.  We believe that the paper now gives the reader the information they need to learn about the irreproducibility problem in deep networks, and about the smooth activations’ approach to reduce this problem.  We believe that with the convincing additional results we provided, the reviewer can adjust their rating of the paper.  The paper presents a problem that is not well known and understood in the literature and the community but is significant and widespread in real world production systems, with a novel solution, and we believe that this is the importance of this paper, on which it should be judged. The merit of the paper and its novelty is in describing the problem (that is not very known in the community), as well as describing an approach that addresses it. We believe that the paper should be accepted and published based on this merit and novelty.
>
> - The **visualization** used to show the surface is similar in pattern to a loss one.  The choice here was made to have a good illustration of how the surface changes.  This can be done with one output or many.  We believe that the graphs shown in the paper demonstrate the message we are trying to convey very well, with no need for additional surfaces.  The 2-d input is a toy example, to allow a 3-d graph.  All graphs use the exact same network, to illustrate the different effects.  The choice of the random network was guided by the one that gives the best illustration of the message we wanted to convey.
>
> - **The choice of beta** is dataset, model architecture, and optimizer dependent.  The paper does not claim to present a method on how to come up with the best choice.  One can try a grid and optimize the parameters according to one’s constraints.  The appendix presents an approach to optimize it for accuracy, but also one to optimize for reproducibility.   The latter requires the optimization for reproducibility to be done separately from training the model for accuracy for the reason that for many values of beta, these are actually conflicting objectives.
>
> - **Multiple optimizers:** The revised version of the paper contains empirical results with three different optimizers: AdaGrad, SGD and Adam.  Results demonstrate that irreproducibility is a problem with all of them, and SmeLU reduces the problem in all cases.  Figure 4 provides results for SGD as well as AdaGrad.
>
> - The datasets used in the original version are at both extremes (huge and small).  The huge dataset demonstrates that the problem can be of high magnitude, while the small one shows that it still occurs in small scales. The Criteo dataset we added also demonstrates the significance of the problem and the benefits for SmeLU.
>
> The paper presents a problem and a possible solution and illustrates them on different datasets.  We added more empirical results, as mentioned.  While it would be nice to show similar results on different datasets and different configurations, the work already demonstrates the problem and the usefulness of the approach to solve it.  We believe that the results presented give sufficient exposure to the problem and the solution approach.

---

### Official Review · AnonReviewer4 · 2020-10-28
**Improving reproducibility with smooth(er) activations that approach relu**

**Rating:** 5
**Confidence:** 3

**Review:**

Summary: This paper addresses the problem that deep neural networks (DNNs) can lead to different predictions (even when they are initialized the same way) due to the stochasticity of samples selected in mini-batch SGD and update procedures from different optimizers, which leads to convergence to different regions along the loss surface.  They attribute this problem to the complicated loss surface that arises from the discontinuity in relu activations. They show that smooth activations can help remedy this issue, by tuning the activation to become more relu-like, which leads to a better tradeoff between prediction differences (i.e. consistency) and model accuracy.

The pitch of the problem is motivated by fields like healthcare where more consistent predictions are important. However, it’s unclear why one would expect reproducible predictions when two network (with same initialization) are trained with a different sampling of mini-batch shuffles. This stochasticity should lead to different solutions for complex surfaces, just as a different initialization would. Smoothing the loss surface can help here and this paper only explores the extent that the smoothness of the activation plays a role. It’s unclear to me whether the activation function alone is sufficient for solving this problem. Other options, such as regularization, are not explored.

Comments:
* It is nice that traditional activations like swish and softplus are parameterized with a beta so that they can be modulated to become more relu-like. Given the stochasticity of mini-batching, it’s not surprising that you can start from the same initialization and navigate to a different region of the loss surface. Since the stochasiticity is due to the size of the mini-batch, this paper could benefit from exploring how the effect size of PDs changes with batch size.
* Overall, the empirical evidence is very light — they compare different activation functions for a private dataset for ad Click-through-rate and MNIST. To make general claims, there should be various networks and various optimizers on various datasets. The paper can be more convincing if more datasets are explored, preferably not on a private dataset for which no one can validate their results.
* Also, the optimizer plays a major role in navigating the loss surface, but only a single optimizer is compared in the first task, while only 2 are explored in task 2 (i.e. adagrad and sgd). Adding more optimizers, especially popular ones like adam, could help clarify whether the smooth activations is robust across optimizers or whether this is a special case for adagrad. I suspect the former, but experiments must be performed to prove any point.
* On a smaller note, what does each dot in Fig 3 represent. Assuming it represents different beta values, there should be more consistency -- softplus has many points while tanhexp only has 4 points. Also, how do these betas alter the activation functions -- the  main/appendix show plots for 5 values of beta. Also, in Fig 4, why aren’t the same betas used for adagrad and sgd?  The choices of beta for a given base activation function should be consistent throughout.
* Fig 4 should also show the standard deviation across the 12 models. This can provide a sense for the statistical significance of the results.
* RESCU, a generalized activation function, is introduced but they never used in any comparisons. What is the benefit of such a formulation? This paper mentions that smelu is less expensive compared to the other smooth activations. Any direct statement like this should be followed up with quantitative comparisons — in this case the time per epoch.
* This paper shows how weight normalization also influences the loss surface, but was not explored empirically. I think adding additional experiments taht scan different weight norms for a fixed beta would only strengthen their claims.
* It is unclear whether their initialization strategy for the DNNs explored here is fixed or whether they simply sampled different weights from the same initialization distribution. The wording of the motivation was that PD happens for the same initialization but the language in the text was ambiguous of whether they enforced the same random number seed for the initializer across each experiment.
* As a control experiment, the effect size of prediction differences between training a model with different initializations should be compared with their main results that explore training with different mini-batch order starting from the same initialization.

---

> ### Author Response · Authors · 2020-11-24
> **Response to AnonReviewer4 Part 1**
>
> We thank the reviewers for their efforts, time and comments, which we respond to below.
>
> The reviewer asks about other approaches to address irreproducibility in deep networks.  We agree with the reviewer that the activation function is not the only cause of prediction differences in deep networks.  However, our focus on **this** paper is on the activation function and on demonstrating that ReLU is a cause of exacerbating prediction differences.  The message in the paper is that this issue can be mitigated by a choice of smooth activation.  There is no attempt in the paper to claim that this is the only cause.  On the contrary, the paper suggests (specifically with the 3-d illustrations) that even with the smooth activations the surface is such that there are still multiple optima, just fewer.  The paper references other works to address irreproducibility.  However, while uncertainty in deep networks received much attention, one of the goals of this paper is to bring irreproducibility to the attention of the community.  While one can classify irreproducibility as a component of uncertainty, it is very different from epistemic model uncertainty, as it actually does not diminish with more training examples.  We are emphasizing these differences more in the paper.  As to the reviewer’s question about other methods - we have explored many different methods to this problem.  We note that many regularization approaches do not actually work to mitigate this problem.
>
> - **Fixing initialization vs. shuffling:** While initialization can be controlled, in huge scale distributed parallel systems training order cannot be fully controlled. If we desire more reproducible predictions, we thus must use other methods.  Results we added for the Criteo dataset also demonstrate that both initialization and shuffling contribute to PD.
>
> - **Batch size:** We agree that the effect of shuffling batch size can be illustrated when studying irreproducibility, but this is beyond the scope we defined for **this** paper.  The scope of this paper is to expose the irreproducibility problem to the community and to describe the smooth activation partial solution.  Our results on Criteo demonstrate the difference between shuffling and no shuffling.  Batch size effect is an orthogonal study on causes of irreproducibility.
>
> - **Benchmark datasets:** The original submission included two extremes on the spectrum of scales - private huge scale datasets, where the problem is substantially more pronounced, on one hand, and the relatively small MNIST dataset, where the problem is less pronounced.  The paper shows that the problem exists in the full spectrum.  We have added results on the Criteo benchmark dataset that clearly demonstrate the problem and the effects of SmeLU in reducing it.  We believe that we now present strong empirical evidence to the problem and the effectiveness of the method we developed to address it. Although we cannot report it, we observed this problem and the effectiveness of SmeLU on many other private datasets.
>
> - **Optimizer:** The choice of optimizers does change the quantitative but not the qualitative behavior.  The revised version presents results with three optimizer: SGD, AdaGrad and Adam.
>
> - **The choice of beta** values depends on the dataset, activation, optimizer, and model architecture.  The values in the results presented were chosen to cover the spectrum of accuracy reproducibility for the specific configuration.  Unfortunately, there is no one size fits all here.  The choice depends on all these factors, and one should tune the desired accuracy/PD tradeoffs when using the activations.
>
> - **Standard deviation:** We added standard deviation on AUC and PD for the Criteo dataset. These even strengthen the benefits of SmeLU, showing significant reductions from ReLU baselines.
>
> - **RESCU** is introduced just as a generalization of the approach.  The Sigmoid-RESCU particular activation seems inferior to SmeLU on the datasets we tested.  The idea in introducing RESCU is to convey the message that the technique to obtain SmeLU and generalized SmeLU is more general and can produce other activations.  There is no attempt to claim that such activations can be better than SmeLU.

---

> > ### Author Response · Authors · 2020-11-24
> > **Response to AnonReviewer4 Part 2**
> >
> > - **Cost:** The claim made for SmeLU cost is that with standard computation, Swish and other smooth activations are more expensive.  However, if internal tables and optimizations are used to compute exponents and logarithms, it is possible that runtime will not be substantially affected.  Systems developed for deep networks have invested effort in optimizing operations like Sigmoid and exponents, so such results may not present the real advantage. Furthermore, measuring runtime on highly parallel modern systems is tricky, as the measurements may be skewed towards the platform implementation away from the experiment. The claim for SmeLU is that if one were to build a specialized hardware to support a smooth activation, it is very likely that the cost would be cheaper to implement linear and quadratic operations as opposed to exponents and logarithms, and such systems could be made to perform faster due to the mathematical simplicity.
> >
> > - **Weight norm:** The results from the real datasets without weight norm were mentioned in the text to be inferior to the results with weight norm. The interaction with weight norm seems to be dataset specific. The effect of normalization, however, is secondary to the message of the paper.
> >
> > - **Initialization:** In a set of M models among which PD was measured, parameter X in all M models was initialized to the same value for real data and MNIST.  This is true for all parameters, and we are clarifying this in the paper.  For Criteo, the results presented in the text initialized each model independently (differently), but we include results in an appendix, in which initialization was identical as well.
> >
> > - **Effect of initialization:** Different initializations can lead to higher PD values: we’ve included such results with the Criteo dataset, and also mentioned results for real data.

---

### Official Review · AnonReviewer1 · 2020-10-29
**Stability of smooth activations as opposed to ReLU**

**Rating:** 4
**Confidence:** 2

**Review:**

The paper claims that smooth activations are more reproducible than ReLU. The accuracy gain claims seem marginal and not carefully carried out, further ablation studies are needed to strengthen the conclusion on accuracy. However, the main point of the paper is reproducibility where the feature is measured by the ‘Prediction Difference’. PD (introduced in section 2) is a measure over a set of models where the PD score is low if the models output consistent estimates for the same validation samples.

Do models with the same initialization, and same randomness seed for the shuffles of SGD have high PD? If so then there would be numerical issues in the way models are trained. Otherwise, where does the difference in PD come from? Why such fluctuations are considered irreproducible? Why would PD be identified with reproducibility?

Overall the paper has a consistent story to tell. It’s a fresh perspective and focuses on the main problem at hand. However, I fail to understand some of the key measurements and their connection with reproducibility. It is undeniable that the shape of the landscape and the consistency of the models are linked. However, the variation found by ReLU’s could help increase ensemble accuracy. In other words, such variation can be a feature depending on the context. And, in principle, given the proper seeds and versions of the software, the results should be fully reproducible. Am I missing something? I am looking forward to the responses from the authors on this point.

---

> ### Author Response · Authors · 2020-11-24
> **Response to AnonReviewer1 part 1**
>
> We thank the reviewer for their efforts, time, comments, and questions.  Below we try to answer some of the reviewer’s questions.  We believe that by addressing these questions, the reviewer would have a better understanding of our paper and our results, and be able to adjust their recommendation.
>
> First, irreproducibility as we describe in this paper is a problem that has had very little exposure in the literature.  It has been often mixed with and aggregated into studies of uncertainty and overfitting.  However, it is a different problem that varies from these other two problems.  While epistemic model uncertainty diminishes with more training data, prediction differences due to irreproducibility do not diminish with more training data.  In some literature, irreproducibility was aggregated into uncertainty studies, because when one measures variances, there is no way to disentangle which portion comes from model parameters’ uncertainty and which from multiple optima in a non-convex objective.  Overfitting hurts prediction accuracy on unseen data by biasing the prediction towards the training data.  Irreproducibility does NOT hurt accuracy.  Irreproducibility in deep networks, as we illustrate in the paper, seems to stem from the fact that the complex non-convex objective space of deep networks has multiple solutions, that seem to have equal average accuracy but lead to very different predictions on individual examples.
>
> The claim in the paper is that smooth activations give a knob that gives better tradeoffs between accuracy and reproducibility.  We believe that this is shown by the results.  While the accuracy improvements are small, they are still there.  (We added a mention in the revised version that even fractions of percent accuracy improvements can be hugely meaningful in huge scale practical systems, such as ones that predict click-through-rate for ads.)  The prediction difference improvements are definitely more substantial and are the focus of the paper.
>
> As for the questions posed by the reviewer:  Different initializations lead to different optima and irreproducibility.  However, even identical initialization leads to different optima and non-diminishing non-negligible prediction differences.  For the Criteo dataset, we now show results with and without identical initializations.  While PD is lower with identical initialization, it can still be very high (for Criteo, over 20% with identical initialization and over 30% with different initializations).  For other results shown in the paper, the initialization of the models among which PD was computed was *identical*.
>
> If one could guarantee complete determinism in training with identical initialization, two models trained with such guarantee should not exhibit prediction differences.  However, in practice this is not the case.  Huge scale systems parallelize and distribute training.  Even taking off the shelf systems like TensorFlow, training will be performed in batches, where examples within the batch are not guaranteed to be visited in the same order every time the same model is trained on the same example set.  Due to distribution of the system with many parallel tasks, even updates on training examples may take different orders.  All these effects can lead two identical models to be locked at different regions of the objective space, each of which with a different optimum, that yields different predictions on the same example.
>
> The reason PD can be considered as a measure of irreproducibility is that once a model is trained and reached one optimum, it will keep its predictions to all examples as explained by this optimum.  An identical model trained on the same data at a different instance will reach a different optimum.  Both models now produce different predictions for the same example, and they are not likely to reach a point of agreement even if one keeps training them with more data.  The notion of irreproducibility here can emerge if one trained model A, and used it to generate predictions.  In a new experiment, one trains model B, identical to A and on the same data, and tries to reproduce the predictions of A.  Due to the prediction differences, these predictions will not be reproduced, and the prediction difference can measure what deviation one could expect between A and B.  If PD is 10%, then, the predictions that B will generate deviate from those of A by 10% of the average prediction to A and B.  High PDs may not be tolerable in systems that rely on the predictions for downstream decisions and actions, that can diverge more widely as they rely on these predictions.

---

> > ### Author Response · Authors · 2020-11-24
> > **Response to AnonReviewer1 Part 2**
> >
> > There are various papers in the literature using ensembles to measure prediction variances.  Smooth activations do not come to replace these approaches.  Such variances can be measured with a SmeLU ensemble as well. However, with SmeLU, this measurement will consist of a larger portion which is epistemic uncertainty, and a smaller component that emerges from the multiple optima.
> >
> > As for using ensembles to improve accuracy - many papers propose the use of ensembles.  However, ensembles require more operations.  If one uses the same number of operations in a single deep network (which is wider, to keep the number of operations fixed), accuracy attained is better than that of the ensemble in huge scale systems.  This is only true in systems for which the single component is not saturated to a point that adding more parameters no longer improves performance accuracy.  For huge scale systems, such saturation is hard to reach.  Using ensembles in such systems thus leverages potential accuracy gains to improve reproducibility, at the cost of accuracy.  Smooth activations allow reducing irreproducibility but also keeping the benefit of a single wide network with better accuracy.

---

### Official Review · AnonReviewer2 · 2020-11-01
**Interesting problem, but metrics and experiments are not convincing**

**Rating:** 2
**Confidence:** 5

**Review:**

This submission studies how the choice of activation function impacts the reproducibility of experiments involving deep networks. It proposes a new activation function with the goal of designing a smoothed ReLU, and provide experiments comparing it against other activations in terms of irreproducibility (measured via PD) and performance.

The problem of understanding how model design choices can have negative impacts on experimental reproducibility is interesting and timely, but I believe the paper does not provide a strong enough case for their approach and contributions.

First, the adopted metric to measure irreproducibility, 'Prediction Difference (PD)', is never evaluated in terms of how sensible of a metric it is to capture reproducibility -- this also seems to be lacking in [1]. Actually , one can argue that it is not a sensible metric at all (except for its Hamming form), as it is not invariant to how the models are calibrated, as discussed below.

For example, take any binary classifier and consider two copies of it with different calibrations (i.e. scaling the output layer weights by positive scalars, one for each model): even though the models always agree on their predicted labels regardless of their calibrations, the PD can be made arbitrarily close to 0.5 by calibrating the models appropriately. Even more worrying is that the same can be done by taking a binary classifier and a copy of it with flipped predictions: the PD between the two can be made arbitrarily close to 0 by scaling their weights down. Note that this problem also happens with the relative PD.

To see how this is connected to the choice of activation functions (especially ReLU x SmeLU), note that for normally-distributed inputs (centered around the origin), gradient's variance of ReLU is 1/4 while for SmeLU it is approximately sigma^2 / (4 beta^2) for large enough beta (sigma^2 being the variance of the input distribution): this discrepancy can have a non-trivial impact on the model's calibration and cause differences in PD to be artifacts.

Since this doesn't happen with the Hamming form of PD I believe Figure 15 in the Appendix to be the most informative one. However, it seems that different activations result in less than 1% prediction discrepancy across models, which is fairly insignificant and hence it is hard to argue that activations actually matter for reproducibility (at least from the presented experiments).

Lastly, it is hard to draw any conclusions from the presented experiments: the CTR results are based on a private dataset while the MNIST ones are extremely small-scale, with both the dataset and the model being arguably toy problems. There are numerous tasks where reproducibility is a prominent issue e.g. deep reinforcement learning, generative modelling (especially GANs), making training a 2-layer network on MNIST a poor choice to evaluate reproducibility problems.

As an additional note, the authors seem to rely heavily on the work of Shamir & Coviello '20 [1] which introduced the PD metric, even though the paper was only made publicly available on arXiv a week --after-- the reviewing period for this submission started. When citing papers which are yet to be made available it would be helpful to introduce and discuss the relevant content in a self-contained way -- while the authors avoided much of my confusion by presenting the full definition of the PD metric, the referred paper has useful information which was not discussed (such as which summand is normalized in its relative form and how the different variants compare).

Since I have major concerns with the paper -- particularly on the reliability of PD as a metric and the unconvincing empirical results -- I am voting for rejection.

[1] Shamir & Coviello, Anti-Distillation: Improving reproducibility of deep networks,


------------


Update after rebuttal:

"It appears that the comment made by the reviewer may stem from an assumption that two models which are compared for PD can be different in the operations they perform to generate the predictions."

This is incorrect, my review does not mention such assumption and my statements hold without it. As stated in my review, I consider models with different weight magnitudes, making no assumptions on the underlying cause.

"PD, as we defined in Section 2, is aimed explicitly at measuring differences between predictions of a set of models that are supposed to be identical in all their components"

Indeed, and my point is that comparing the PD of two sets of models that are not identical is also problematic **even if all models within each set are identical**, except for the PD in its Hamming form. More details below.

"Changing calibration between such models violates this assumption."

Please check the celebrated work of Guo et al., "On Calibration of Modern Neural Networks": calibration does not necessarily consist of an explicit, additional component that modifies the model, and the same model trained in different ways can present distinct calibrations. More specifically, two sets of models can have not only the same accuracy, but the exact same predictions (i.e. there is a 1-1 mapping from each model in one set to a model in the other set that has the exact same predictions for all data points) but vastly different internal calibrations, which will result in vastly different PDs (to be overly specific, the scalar PD of a set will be different from the scalar PD of the other set) even though the two sets agree "point-wise" in terms of predictions.

"If one changes something about one of the models (including how calibration is done), one would expect them to predict differently, and have different accuracies."

This is incorrect. First, I'm not assuming models are explicitly calibrated, only that they have distinct internal calibrations (confidences in terms of predicted probabilities, which depend mostly on the parameters' magnitudes). Second, "scaling the output layer weights by positive scalars" (quoting from my review) will not change a model's accuracy: while it changes the class-wise predicted probabilities, the rank of the logits is preserved. If the authors remain skeptical of this fact, let $\phi(x)$ denote the activations of the previous to last layer of a model, and let $\langle w_i, \phi(x) \rangle > \langle w_j, \phi(x) \rangle$, where $w_i$ and $w_j$ are the weight vectors of output units respective to classes $i$ and $j$ (i.e. $p(y_i | x) > p(y_j | x)$ for probabilities produced by a softmax over logits). Then for any $\alpha \in \mathbb R_+$, we have trivially that $\langle \alpha w_i, \phi(x) \rangle > \langle \alpha w_j, \phi(x) \rangle$ (hence $p'(y_i | x) > p'(y_j | x)$, for probabilities $p'$ computed from the new logits). Again, note that it is --not-- necessary for an external, explicit calibration factor $\alpha$ to be employed: training the network differently, or even adopting a different activation function -- just consider $\max(0, 10x)$ for clarity, which will scale $\phi(x)$ by a positive factor and yield the same observation as above.

"Specifically, if one flips the predictions of a binary classifier, the flipped model will have much worse accuracy from the actual model of interest, and measuring PD at this point is irrelevant."

The fact that two classifiers with vastly different accuracies can have zero PD is worrying and shows that PD is not a trustworthy metric: claiming that such evaluation is 'irrelevant' and should not be done does not address the issue.


Since the authors remained unconvinced that the PD is sensible to positive scalings of a model's parameters, and hence comparing the PDs of two sets of models with different activations (one activation per set) is not sensible, here is a more detailed explanation of this fact.

Assume a fairly trivial example for clarity: two 1-d data points, $x_1 = +1, x_2 = -1$, and binary classification models $f_1, f_2$, where $f_1(x) = \sigma(w_1 \cdot \phi(x))$ and $f_2(x) = \sigma(w_2 \cdot \phi(x))$ are the assigned probabilities for the positive label, and $\phi: \mathbb R \to \mathbb R$ captures some notion of activation function and/or scale of weights before the final classification layer. For simplicity, let $\phi(x) = \alpha x$, for some $\alpha \in \mathbb R_+$, and feel free to think of $\alpha$ as a 'magnitude' of an activation function instead of some notion of internal calibration.

Then, we have $P_{1,1} = (\sigma(\alpha w_1), \sigma(-\alpha w_1))$, $P_{1,2} = (\sigma(\alpha w_2), \sigma(-\alpha w_2))$, $P_{2,1} = (\sigma(-\alpha w_1), \sigma(\alpha w_1))$, and $P_{2,2} = (\sigma(-\alpha w_2), \sigma(\alpha w_2))$. The PD of the set consisting of the two defined models, after simplifying the 8 relevant terms, ends up being simply $\Delta_1 = |\sigma(\alpha w_1) - \sigma(\alpha w_2)|$. Let's pick some numbers to make this crystal clear: let $\alpha = 1, w_1 = 1.0, w_2 = 0.1$, so we get $\Delta_1 = \sigma(1) - \sigma(0.1) \approx 0.2$ (note that w.l.o.g. we can assume that $y_1 = +1, y_2 = -1$ so that for these weights both models achieve 100% accuracy).

Now, take ANOTHER set, consisting of models $g_1, g_2$, defined similarly to $f_1, f_2$, but with $g_1(x) = \sigma(w'_1 \cdot \phi'(x)), g_2(x) = \sigma(w'_2 \cdot \phi'(x))$, where $\phi'$ (not the derivative of $\phi$) captures the the activation function and/or weight magnitude of layers preceding the classification head. Let $\phi'(x) = \beta x$ for simplicity. Consider the case where $\beta = 0.1, w_1 = 1.0, w_2 = 0.1$, i.e. the weights of $g_1, g_2$ are *exactly the same* as the weights of $f_1, f_2$, but $\phi'$ is a 'scaled-down' $\phi$ (e.g. a different activation function): in this case (note that both $g_1$ and $g_2$ achieve 100% accuracy as well), **for this new set of models, consisting of the pair $g_1, g_2$**, we get $\Delta_1 = \sigma(0.1) - \sigma(0.01) \approx 0.02$, a value around 10 times smaller than the PD of the first set of models, **even though the second set predicts the exact same labels for each data point**, and claiming that the set $\{g_1, g_2\}$ is 'more robust' than the set $\{f_1, f_2\}$ in terms of reproducibility is simply factually wrong. If the idea of having $\beta \neq \alpha$ sounds a bit of a stretch since the proposed activations are not simply 'scaled down' ReLUs, consider instead the case $\beta = 1.0, w_1 = 0.1, w_2 = 0.01$ and note that we again get $\Delta_1 \approx 0.02$ for this second set of models: the discrepancy in terms of magnitude of weights can be caused by different optimizers, different strength of $\ell_2$ regularization, or, as my original review already mentioned, smaller variance of gradients w.r.t. activation function.

To reiterate, in the above example we did **not**, at any point, compute the PD of a set of models that had different components: both $\{f_1, f_2\}$ (the first set) had the same 'activation function' $\phi$, while $\{g_1, g_2\}$ had $\phi'$.

Going a step further, which shows how problematic the PD is as a metric, consider an arbitrary set of binary classifiers $S_1 = \{f_1, f_2, \dots, f_M\}$, where $f_i(x) = \sigma( \langle w_i, \phi(x) \rangle)$ is the probability assigned by the $i$'th model of $x$ belonging to the positive class. Now, take *another* set of binary classifiers $S_2 = \{g_1, g_2, \dots, g_M\}$, with $g_i(x) = \sigma(\langle w_i, \phi'(x) \rangle)$, where $w_i$ is the **same** weight vector that model $f_i$ has (i.e. except for $\phi'$, the set $S_2$ is 'point-wise' identical to the set $S_1$). Finally, let $\phi'(x) = \beta \phi(x)$, where $\beta \in \mathbb R_+$, and feel free to check that for any $\beta$, every model $g_i$ from $S_2$ will agree with the model $f_i$ from $S_1$ in terms of predicted class (i.e. although the class probabilities will change, the rank is be preserved for any $\beta$). This means that $S_2$ produces the **exact same** predictions as $S_1$ for **any possible data point**. Taking $\beta \to 0$ yields in $g_i(x) \to 0.5$ for any $i \in [M]$ and possible $x$, hence **the PD of $S_2$ will go to zero, even though the PD of $S_1$ can be arbitrarily large and the two model sets $S_1, S_2$ agree point-wise in terms of predicted classes**. In other words, taking an arbitrary set of models with ReLU activations, copying its weights and replacing the ReLU by $\phi(x) = \max(0, \frac{x}{10^{10}})$, will yield a second model set with PD close to zero. Hopefully the authors agree with me that this trivial replacement of activation functions does not 'solve' any reproducibility problem in machine learning.

With the above in mind, I urge the authors to re-evaluate PD as a metric. As mentioned in my review, the Hamming form does not suffer from this issue, but the reported numbers in this case seem to indicate that there is little to no reproducibility challenge for the adopted tasks.

---

> ### Author Response · Authors · 2020-11-24
> **Response to AnonReviewer2**
>
> Due the character limit, we post the response as two comments.
>
> We thank the reviewer for their effort, time, and detailed comments.  However, we believe that the reviewer’s recommendation stems from misconceptions about the merit of the prediction difference metric, and from their belief that the experimental results are not sufficient to illustrate the problem and that smooth activations address this problem.  We hope that our revised version and response below gives the reviewer sufficient information to understand that the PD metric is a good choice to measure irreproducibility we care about, and that the datasets we provided, including the additional results on Criteo, do demonstrate the problem and the advantages of smooth activations and SmeLU.  We hope that with this information, the reviewer is able to re-evaluate their rating of the paper.  As we state below, we observed the results we reported on various private datasets with different models and different scales.  We believe that the private dataset demonstrates the realistic problem with very substantive numbers on PD.  It is important to present these results to demonstrate to the community the real practical importance of this problem.  Unfortunately, we are unable to disclose more information about the real dataset, and while benchmark datasets are common in this literature, their scale is substantially lower than that of real systems. The added results on the Criteo dataset, however, demonstrate similar magnitudes of the problem. Since MNIST is a much smaller dataset, the magnitude of this phenomenon is much less substantial than in real systems.
>
> The paper studies irreproducibility in predictions of deep models that are supposed to be identical.  The focus is to provide a solution for practical real world systems that suffer from irreproducibility and are of large or huge scale.  The goal is not to design a smooth ReLU, but to design a method that mitigates prediction differences, at no cost to accuracy and that uses a least expensive and least complex approach.
>
> We thank the reviewer for the comments about the prediction difference (PD) metrics.  However, we respectfully disagree with the claims discrediting this metric.  Our paper defined irreproducibility as the fact that predictions on individual examples differ between supposedly identical models.  PD measures exactly those differences.
>
> It appears that the comment made by the reviewer may stem from an assumption that two models which are compared for PD can be different in the operations they perform to generate the predictions.  This is not the case in the paper.  PD, as we defined in Section 2, is aimed explicitly at measuring differences between predictions of a set of models that are supposed to be identical in all their components, including any calibration they perform to produce some final score. Changing calibration between such models violates this assumption.  Furthermore, PD is only of interest when the models in the set have roughly the same average accuracy over the test data.  The metric is not intended to compare between models that are different in any sense.  The irreproducibility problem, as explained in the paper, occurs when two models that are supposedly identical, and trained on the exact same set of training examples, give different predictions, but have the same average accuracy.  If one changes something about one of the models (including how calibration is done), one would expect them to predict differently, and have different accuracies.  Specifically, if one flips the predictions of a binary classifier, the flipped model will have much worse accuracy from the actual model of interest, and measuring PD at this point is irrelevant.  We further emphasize that PD on prediction probability is measured on the actual final probability value predicted for the label, not just on ranking.  Calibrating two models differently will change this score, but again, this is irrelevant to the purpose of the PD metric we defined.  Furthermore, one can make PD zero by predicting 0.5 probability for a positive label for every example.  However, such a model will have very bad accuracy.  The aim here is to keep good accuracy, and mitigate the differences in predictions, which is what is achieved with smooth activations.

---

> > ### Author Response · Authors · 2020-11-24
> > **Response to AnonReviewer2 part 2**
> >
> > This is the second part of the response.
> >
> > The discussion in Shamir & Coviello adds some more insight why this choice of metric is used as opposed to KL divergence or other approaches.  However, we included all the relevant information to the metrics we chose to use in this paper in a self-contained way.  We believe that the information in this paper is sufficient for the reader’s needs of a self contained exposition, subject to the page limits.  We believe that the parts in Shamir & Coviello that were not included are not required for the understanding of this paper.
> >
> > By the explanations above, we do not find the reviewer’s discussion about gradient variances relevant to the work in this paper.  We never measure PD between a SmeLU and ReLU activated model.  We measure PD within a set of SmeLU models and within a set of ReLU models, and compare those PD and accuracy values between the two sets.
> >
> > The PD numbers for the small MNIST dataset are smaller.  However, with the large scale real data, we observe substantial PD values of over 10% (and over 20% and 30% for Critoe).  We believe that the scale of the problem contributes to the amount of PD measured, and have seen even larger numbers with even larger scale problems, which due to data ownership, we are unable to report publicly.
> >
> > While the reviewer claims that only the Hamming metrics should be trusted, we respectfully disagree with this claim.  Furthermore, we have observed that all metrics behave rather similarly, and the graphs obtained for different values of the parameters were very similar.
> >
> > The purpose of initially reporting two extremes (huge scale real data and MNIST) was to show that the problem exists in deep networks across a full spectrum of problem scales, and is mitigated to scale by the use of smooth activations.  We observed the problem in many different private datasets with different models and different scales.  While we believe that the material already in the paper clearly demonstrates this problem and that smooth activations are able to mitigate it, we integrated additional results on the Criteo dataset that clearly demonstrate the problem and the solution approach as well.  We believe, however, that the large scale results demonstrate the problem clearly, and they show that this is a problem in reality.  Benchmark datasets are much smaller, and may not capture phenomena that are observed in real world huge scale problems as well.  However, even with the small MNIST dataset, we observe behavior that is consistent with the huge scale, only at a smaller scale.

---

> ### Author Response · Authors · 2021-02-13
> **Authors response to post rebuttal comments made by the reviewer.**
>
> We thank the reviewer for the thorough description of their concern.  However, our paper focuses on problems where the actual value of the prediction score and prediction loss matter.  The Hamming metric is presented as an alternative for problems when one only cares about the correctness of the predicted label.  We thus respectfully disagree with the claims made by the reviewer post the rebuttal stage, which we believe are irrelevant

---

### Author Response · Authors · 2020-11-24
**Response to all reviewers and area chairs**

We thank the reviewers for the time they spend on this paper, the efforts they made to give us feedback, and their detailed comments.  We made an effort to address these comments in the revised version.

Going thoroughly through the reviews we believe that the only major concern is of limited experimental benchmark datasets results.   We addressed that by adding results on the Criteo dataset that demonstrate very clearly that irreproducibility is a problem in deep networks, and that SmeLU can reduce the magnitude of prediction differences, as well as variance of accuracy metrics and of the prediction difference itself.  In our results, we clearly show that smooth activations give better reproducibility with equal or better accuracy.  While one can always add more benchmarks to demonstrate a problem and the proposed solution, we believe that all the results we presented in the revised version now very clearly illustrate both the problem we are addressing and the strength of the proposed solution.  More results would always be nice to have, but we do not agree that the publication of the paper should be delayed because of this issue, especially with the strong additional results we added to the revised version.

We believe that the paper describes a very important problem, that has been given very little attention in the literature, and gives a novel solution to mitigate this problem.  While in hindsight, using smooth activations is a good solution to reduce irreproducibility, it is an approach that is very hard to conclude and come up with when observing the irreproducibility problem.  As such, we think that the merit of this paper is in both describing a problem that has been neglected from most deep networks literature, and describing a novel approach to address it.  The community can benefit from both; learning about this problem, and seeing a cheap method to address it.  We do provide clear evidence to this problem and to the benefit from smooth activations. We thus believe that rejecting this paper on the grounds of not enough empirical evidence is not a correct decision, especially as we believe we now provided sufficient empirical evidence. We believe and claim that the contributions in this paper are already solid and justified by the existing content, and it thus should be accepted and published.  We appreciate the consideration, and hope that you understand and agree with our claims, with the additional evidence we have provided.

---

### Author Response · Authors · 2021-02-13
**Authors Response to Final Decision**

The final decision to reject the paper is based solely on the claim that irreproducibility was not demonstrated on multiple benchmark datasets beyond the ones presented.  We respectfully disagree with this argument.  As several recent works demonstrate, irreproducibility (underspecification, nondeterminism) is indeed a critical problem that affects many practical problems.  Unfortunately, some reviewers are unaware of the problem.  However, this does not make the problem less critical, especially in real world practical systems, that are at a scale much larger than the “toy” benchmark datasets presented in the literature.  This is the reason it was important to present results on a private large scale dataset.  See, for example, these recent papers written by independent groups to ours to realize the importance of this problem:

- D’Amour et. al., Underspecification presents challenges for credibility in modern machine learning
- Summers and Dinneen, On nondeterminism and instability in neural network optimization.

Our paper presented a novel look and approach to mitigate this problem.  We do believe that we demonstrated sufficient evidence to its existence, and to the effect of the technique we presented to reduce the amount of irreproducibility observed.  We respectfully disagree with the final decision, as we believe that the community will come to realize the importance of this problem and of techniques to address it in the coming years.  We also respectfully disagree with the claim that “differences in performance are not significant”, they are clearly significant for reproducibility.  Finally, the post rebuttal claims by AnonReviewer2 clarify that the reviewer considers only classification when one only cares about whether the correct label was predicted.  Our exposition focuses on problems in which the prediction score itself (and objective/loss measured) are important.  We defined prediction difference and loss/objective metrics on these values, and not on the final label prediction only.  We argue that this case is the important one in very many practical problems, and even in multiclass classification, when one needs a sufficiently large score to make a decision, instead of declaring that no such decision could be made.  While the Hamming version of PD applies to the final label prediction, the other metrics apply to the actual prediction value, and trade off with accuracy measured by prediction loss (and not only by the final predicted label accuracy).

---

### Decision · Program_Chairs · 2021-01-07
**Final Decision**

**Decision:**

Reject

**Comment:**

All reviews were negative for this paper, due to various issues. I think the main issue was that the experimental results were too weak to be convincing. For example, the reviewers were not sure if the differences in performance between different activations are significant. The reviewers also required more datasets and more experiments. The authors added std to results, more experiments and argued that the current datasets are sufficient, but the reviewers seemed to remain unconvinced.